# Variational Bayesian Optimal Experimental Design

**Adam Foster**[†*]  **Martin Jankowiak**[‡]   **Eli Bingham**[‡]   **Paul Horsfall**[‡]
**Yee Whye Teh**[†]   **Tom Rainforth**[†]   **Noah Goodman**[‡§]
[†]Department of Statistics, University of Oxford, Oxford, UK
[‡]Uber AI Labs, Uber Technologies Inc., San Francisco, CA, USA
[§]Stanford University, Stanford, CA, USA
adam.foster@stats.ox.ac.uk

## Abstract

Bayesian optimal experimental design (BOED) is a principled framework for making efficient use of limited experimental resources. Unfortunately, its applicability is hampered by the difficulty of obtaining accurate estimates of the expected information gain (EIG) of an experiment. To address this, we introduce several classes of fast EIG estimators by building on ideas from amortized variational inference. We show theoretically and empirically that these estimators can provide significant gains in speed and accuracy over previous approaches. We further demonstrate the practicality of our approach on a number of end-to-end experiments.

## 1   Introduction

Tasks as seemingly diverse as designing a study to elucidate human cognition, selecting the next query point in an active learning loop, and designing online feedback surveys all constitute the same underlying problem: designing an experiment to maximize the information gathered. Bayesian optimal experimental design (BOED) forms a powerful mathematical abstraction for tackling such problems [8, 23, 37, 43] and has been successfully applied in numerous settings, including psychology [30], Bayesian optimization [16], active learning [15], bioinformatics [42], and neuroscience [38].

In the BOED framework, we construct a predictive model $p(y|\theta, d)$ for possible experimental outcomes $y$, given a design $d$ and a particular value of the parameters of interest $\theta$. We then choose the design that optimizes the expected information gain (EIG) in $\theta$ from running the experiment,

$$\text{EIG}(d) \triangleq \mathbb{E}_{p(y|d)}\big[H[p(\theta)] - H[p(\theta|y, d)]\big], \tag{1}$$

where $H[\cdot]$ represents the entropy and $p(\theta|y, d) \propto p(\theta)p(y|\theta, d)$ is the posterior resulting from running the experiment with design $d$ and observing outcome $y$. In other words, we seek the design that, in expectation over possible experimental outcomes, most reduces the entropy of the posterior over our target latent variables. If the predictive model is correct, this forms a design strategy that is (one-step) optimal from an information-theoretic viewpoint [24, 37].

The BOED framework is particularly powerful in sequential contexts, where it allows the results of previous experiments to be used in guiding the designs for future experiments. For example, as we ask a participant a series of questions in a psychology trial, we can use the information gathered from previous responses to ask more pertinent questions in the future, that will, in turn, return more information. This ability to design experiments that are self-adaptive can substantially increase their efficiency: fewer iterations are required to uncover the same level of information.

In practice, however, the BOED approach is often hampered by the difficulty of obtaining fast and high-quality estimates of the EIG: due to the intractability of the posterior $p(\theta|y, d)$, it constitutes

a nested expectation problem and so conventional Monte Carlo (MC) estimation methods cannot be applied [33]. Moreover, existing methods for tackling nested expectations have, in general, far inferior convergence rates than those for conventional expectations [22, 30, 32]. For example, nested MC (NMC) can only achieve, at best, a rate of $\mathcal{O}(T^{-1/3})$ in the total computational cost $T$ [33], compared with $\mathcal{O}(T^{-1/2})$ for conventional MC.

To address this, we propose a variational BOED approach that sidesteps the double intractability of the EIG in a principled manner and yields estimators with convergence rates in line with those for conventional estimation problems. To this end, we introduce four efficient and widely applicable variational estimators for the EIG. The different methods each present distinct advantages. For example, two allow training with implicit likelihood models, while one allows for asymptotic consistency even when the variational family does not contain the target distribution.

We theoretically confirm the advantages of our estimators, showing that they all have a convergence rate of $\mathcal{O}(T^{-1/2})$ when the variational family contains the target distribution. We further verify their practical utility using a number of experiment design problems inspired by applications from science and industry, showing that they provide significant empirical gains in EIG estimation over previous methods and that these gains lead, in turn, to improved end-to-end performance.

To maximize the space of potential applications and users for our estimators, we provide[2] a general-purpose implementation of them in the probabilistic programming system Pyro [5], exploiting Pyro's first-class support for neural networks and variational methods.

## 2 Background

The BOED framework is a model-based approach for choosing an experiment design $d$ in a manner that optimizes the information gained about some parameters of interest $\theta$ from the outcome $y$ of the experiment. For instance, we may wish to choose the question $d$ in a psychology trial to maximize the information gained about an underlying psychological property of the participant $\theta$ from their answer $y$ to the question. In general, we adopt a Bayesian modelling framework with a prior $p(\theta)$ and a predictive model $p(y|\theta, d)$. The information gained about $\theta$ from running experiment $d$ and observing $y$ is the reduction in entropy from the prior to the posterior:

$$\text{IG}(y, d) = H[p(\theta)] - H[p(\theta|y, d)]. \tag{2}$$

At the point of choosing $d$, however, we are uncertain about the outcome. Thus, in order to define a metric to assess the utility of the design $d$ we take the expectation of $\text{IG}(y, d)$ under the marginal distribution over outcomes $p(y|d) = \mathbb{E}_{p(\theta)}[p(y|\theta, d)]$ as per (1). We can further rearrange this as

$$\text{EIG}(d) = \mathbb{E}_{p(y,\theta|d)}\left[\log \frac{p(\theta|y, d)}{p(\theta)}\right] = \mathbb{E}_{p(y,\theta|d)}\left[\log \frac{p(y, \theta|d)}{p(\theta)p(y|d)}\right] = \mathbb{E}_{p(y,\theta|d)}\left[\log \frac{p(y|\theta, d)}{p(y|d)}\right] \tag{3}$$

with the result that the EIG can also be interpreted as the mutual information between $\theta$ and $y$ given $d$, or the epistemic uncertainty in $y$ averaged over the prior $p(\theta)$. The Bayesian optimal design is defined as $d^* \triangleq \arg\max_{d \in \mathcal{D}} \text{EIG}(d)$, where $\mathcal{D}$ is the set of permissible designs.

Computing the EIG is challenging since neither $p(\theta|y, d)$ or $p(y|d)$ can, in general, be found in closed form. Consequently, the integrand is intractable and conventional MC methods are not applicable. One common way of getting around this is to employ a nested MC (NMC) estimator [30, 43]

$$\hat{\mu}_{\text{NMC}}(d) \triangleq \frac{1}{N}\sum_{n=1}^{N} \log \frac{p(y_n|\theta_{n,0}, d)}{\frac{1}{M}\sum_{m=1}^{M} p(y_n|\theta_{n,m}, d)} \quad \text{where} \quad \theta_{n,m} \overset{\text{i.i.d.}}{\sim} p(\theta), \ y_n \sim p(y|\theta = \theta_{n,0}, d). \tag{4}$$

Rainforth et al. [33] showed that this estimator, which has a total computational cost $T = \mathcal{O}(NM)$, is consistent in the limit $N, M \to \infty$ with RMSE convergence rate $\mathcal{O}(N^{-1/2} + M^{-1})$, and that it is asymptotically optimal to set $M \propto \sqrt{N}$, yielding an overall rate of $\mathcal{O}(T^{-1/3})$.

Given a base EIG estimator, a variety of different methods can be used for the subsequent optimization over designs, including some specifically developed for BOED [1, 29, 32]. In our experiments, we

will adopt Bayesian optimization [39], due to its sample efficiency, robustness to multi-modality, and ability to deal naturally with noisy objective evaluations. However, we emphasize that our focus is on the base EIG estimator and that our estimators can be used more generally with different optimizers.

The static design setting we have implicitly assumed thus far in our discussion can be generalized to sequential contexts, in which we design $T$ experiments $d_1, ..., d_T$ with outcomes $y_1, ..., y_T$. We assume experiment outcomes are conditionally independent given the latent variables and designs, i.e.

$$p(y_{1:T}, \theta | d_{1:T}) = p(\theta) \prod_{t=1}^{T} p(y_t | \theta, d_t). \qquad (5)$$

Having conducted experiments $1, ..., t-1$, we can design $d_t$ by incorporating data in the standard Bayesian fashion: at experiment iteration $t$, we replace the prior $p(\theta)$ in (3) with $p(\theta | d_{1:t-1}, y_{1:t-1})$, the posterior conditional on the first $t-1$ designs and outcomes. We can thus conduct an adaptive sequential experiment in which we optimize the choice of the design $d_t$ at each iteration.

## 3   Variational Estimators

Though consistent, the convergence rate of the NMC estimator is prohibitively slow for many practical problems. As such, EIG estimation often becomes the bottleneck for BOED, particularly in sequential experiments where the BOED calculations must be fast enough to operate in real-time.

In this section we show how ideas from amortized variational inference [10, 17, 34, 40] can be used to sidestep the double intractability of the EIG, yielding estimators with much faster convergence rates thereby alleviating the EIG bottleneck. A key insight for realizing why such fundamental gains can be made is that the NMC estimator is inefficient because a *separate* estimate of the integrand in (3) is made for each $y_n$. The variational approaches we introduce instead look to directly learn a *functional approximation*—for example, an approximation of $y \mapsto p(y|d)$—and then evaluate this approximation at multiple points to estimate the integral, thereby allowing information to be shared across different values of $y$. If $M$ evaluations are made in learning the approximation, the total computational cost is now $T = \mathcal{O}(N + M)$, yielding substantially improved convergence rates.

**Variational posterior $\hat{\mu}_{\text{post}}$**   Our first approach, which we refer to as the variational posterior estimator $\hat{\mu}_{\text{post}}$, is based on learning an amortized approximation $q_p(\theta|y, d)$ to the posterior $p(\theta|y, d)$ and then using this to estimate the EIG:

$$\text{EIG}(d) \approx \mathcal{L}_{\text{post}}(d) \triangleq \mathbb{E}_{p(y,\theta|d)} \left[ \log \frac{q_p(\theta|y, d)}{p(\theta)} \right] \approx \hat{\mu}_{\text{post}}(d) \triangleq \frac{1}{N} \sum_{n=1}^{N} \log \frac{q_p(\theta_n | y_n, d)}{p(\theta_n)}, \quad (6)$$

where $y_n, \theta_n \overset{\text{i.i.d.}}{\sim} p(y, \theta | d)$ and $\hat{\mu}_{\text{post}}(d)$ is a MC estimator of $\mathcal{L}_{\text{post}}(d)$. We draw samples of $p(y, \theta | d)$ by sampling $\theta \sim p(\theta)$ and then $y|\theta \sim p(y|\theta, d)$. We can think of this approach as amortizing the cost of the inner expectation, instead of running inference separately for each $y$.

To learn a suitable $q_p(\theta|y, d)$, we show in Appendix A that $\mathcal{L}_{\text{post}}(d)$ forms a variational lower bound $\text{EIG}(d) \geq \mathcal{L}_{\text{post}}(d)$ that is tight if and only if $q_p(\theta|y, d) = p(\theta|y, d)$. Barber and Agakov [3] used this bound to estimate mutual information in the context of transmission over noisy channels, but the connection to experiment design has not previously been made.

This result means we can learn $q_p(\theta|y, d)$ by introducing a family of variational distributions $q_p(\theta|y, d, \phi)$ parameterized by $\phi$ and then maximizing the bound with respect to $\phi$:

$$\phi^* = \arg \max_{\phi} \mathbb{E}_{p(y,\theta|d)} \left[ \log \frac{q_p(\theta|y, d, \phi)}{p(\theta)} \right], \qquad \text{EIG}(d) \approx \mathcal{L}_{\text{post}}(d; \phi^*). \qquad (7)$$

Provided that we can generate samples from the model, this maximization can be performed using stochastic gradient methods [35] and the unbiased gradient estimator

$$\nabla_{\phi} \mathcal{L}_{\text{post}}(d; \phi) \approx \frac{1}{S} \sum_{i=1}^{S} \nabla_{\phi} \log q_p(\theta_i | y_i, d, \phi) \quad \text{where} \quad y_i, \theta_i \overset{\text{i.i.d.}}{\sim} p(y, \theta | d), \qquad (8)$$

and we note that no reparameterization is required as $p(y, \theta | d)$ is independent of $\phi$. After $K$ gradient steps we obtain variational parameters $\phi_K$ that approximate $\phi^*$, which we use to compute

a corresponding EIG estimator by constructing a MC estimator for $\mathcal{L}_{\text{post}}(d; \phi)$ as per (6) with $q_p(\theta_n|y_n, d) = q_p(\theta_n|y_n, d, \phi_K)$. Interestingly, the tightness of $\mathcal{L}_{\text{post}}(d)$ turns out to be equal to the expected *forward* KL divergence[3] $\mathbb{E}_{p(y|d)}\left[\text{KL}\left(p(\theta|y, d)||q_p(\theta|y, d, \phi)\right)\right]$ so we can view this approach as learning an amortized proposal by minimizing this expected KL divergence.

**Variational marginal $\hat{\mu}_{\text{marg}}$** In some scenarios, $\theta$ may be high-dimensional, making it difficult to train a good variational posterior approximation. An alternative approach that can be attractive in such cases is to instead learn an approximation $q_m(y|d)$ to the marginal density $p(y|d)$ and substitute this into the final form of the EIG in (3). As shown in Appendix A, this yields an *upper bound*

$$\text{EIG}(d) \leq \mathcal{U}_{\text{marg}}(d) \triangleq \mathbb{E}_{p(y, \theta|d)}\left[\log \frac{p(y|\theta, d)}{q_m(y|d)}\right] \approx \hat{\mu}_{\text{marg}}(d) \triangleq \frac{1}{N}\sum_{n=1}^{N}\log\frac{p(y_n|\theta_n, d)}{q_m(y_n|d)}, \quad (9)$$

where again $y_n, \theta_n \overset{\text{i.i.d.}}{\sim} p(y, \theta|d)$ and the bound is tight when $q_m(y|d) = p(y|d)$. Analogously to $\hat{\mu}_{\text{post}}$, we can learn $q_m(y|d)$ by introducing a variational family $q_m(y|d, \phi)$ and then performing stochastic gradient descent to *minimize* $\mathcal{U}_{\text{marg}}(d, \phi)$. As with $\hat{\mu}_{\text{post}}$, this bound was studied in a mutual information context [31], but it has not been utilized for BOED before.

**Variational NMC $\hat{\mu}_{\text{VNMC}}$** As we will show in Section 4, $\hat{\mu}_{\text{post}}$ and $\hat{\mu}_{\text{marg}}$ can provide substantially faster convergence rates than NMC. However, this comes at the cost of converging towards a biased estimate if the variational family does not contain the target distribution. To address this, we propose another EIG estimator, $\hat{\mu}_{\text{VNMC}}$, which allows one to trade-off resources between the fast learning of a biased estimator permitted by variational approaches, and the ability of NMC to eliminate this bias.[4]

We can think of the NMC estimator as approximating $p(y|d)$ using $M$ samples from the prior. At a high-level, $\hat{\mu}_{\text{VNMC}}$ is based around learning a proposal $q_v(\theta|y, d)$ and then using samples from this proposal to make an importance sampling estimate of $p(y|d)$, potentially requiring far fewer samples than NMC. Formally, it is based around a bound that can be arbitrarily tightened, namely

$$\text{EIG}(d) \leq \mathbb{E}\left[\log p(y|\theta_0, d) - \log \frac{1}{L}\sum_{\ell=1}^{L}\frac{p(y, \theta_\ell|d)}{q_v(\theta_\ell|y, d)}\right] \triangleq \mathcal{U}_{\text{VNMC}}(d, L) \quad (10)$$

where the expectation is taken over $y, \theta_{0:L} \sim p(y, \theta_0|d) \prod_{\ell=1}^{L} q_v(\theta_\ell|y, d)$, which corresponds to one sample $y, \theta_0$ from the model and $L$ samples from the approximate posterior conditioned on $y$. To the best of our knowledge, this bound has not previously been studied in the literature. As with $\hat{\mu}_{\text{post}}$ and $\hat{\mu}_{\text{marg}}$, we can minimize this bound to train a variational approximation $q_v(\theta|y, d, \phi)$. Important features of $\mathcal{U}_{\text{VNMC}}(d, L)$ are summarized in the following lemma; see Appendix A for the proof.

**Lemma 1.** *For any given model $p(\theta)p(y|\theta, d)$ and valid $q_v(\theta|y, d)$,*

1. $\text{EIG}(d) = \lim_{L\to\infty}\mathcal{U}_{VNMC}(d, L) \leq \mathcal{U}_{VNMC}(d, L_2) \leq \mathcal{U}_{VNMC}(d, L_1) \quad \forall L_2 \geq L_1 \geq 1,$

2. $\mathcal{U}_{VNMC}(d, L) = \text{EIG}(d) \quad \forall L \geq 1 \quad if \quad q_v(\theta|y, d) = p(\theta|y, d) \quad \forall y, \theta,$

3. $\mathcal{U}_{VNMC}(d, L) - \text{EIG}(d) = \mathbb{E}_{p(y|d)}\left[\text{KL}\left(\prod_{\ell=1}^{L} q_v(\theta_\ell|y, d) || \frac{1}{L}\sum_{\ell=1}^{L} p(\theta_\ell|y, d)\prod_{k\neq\ell} q_v(\theta_k|y, d)\right)\right]$

Like the previous bounds, the VNMC bound is tight when $q_v(\theta|y, d) = p(\theta|y, d)$. Importantly, the bound is also tight as $L \to \infty$, even for imperfect $q_v$. This means we can obtain asymptotically unbiased EIG estimates even when the true posterior is not contained in the variational family.

Specifically, we first train $\phi$ using $K$ steps of stochastic gradient on $\mathcal{U}_{\text{VNMC}}(d, L)$ with some fixed $L$. To form a final EIG estimator, however, we use a MC estimator of $\mathcal{U}_{\text{VNMC}}(d, M)$ where typically $M \gg L$. This final estimator is a NMC estimator that is consistent as $N, M \to \infty$ with $\phi_K$ fixed

$$\hat{\mu}_{\text{VNMC}}(d) \triangleq \frac{1}{N}\sum_{n=1}^{N}\left(\log p(y_n|\theta_{n,0}, d) - \log\frac{1}{M}\sum_{m=1}^{M}\frac{p(y_n, \theta_{n,m}|d)}{q_v(\theta_{n,m}|y_n, d, \phi_K)}\right) \quad (11)$$

where $\theta_{n,0} \overset{\text{i.i.d.}}{\sim} p(\theta)$, $y_n \sim p(y|\theta = \theta_{n,0}, d)$ and $\theta_{n,m} \sim q_v(\theta|y = y_n, d, \phi_K)$. In practice, performance is greatly enhanced when the proposal $q_v$ is a good, if inexact, approximation to the posterior. This significantly improves upon traditional $\hat{\mu}_{\text{NMC}}$, which sets $q_v(\theta|y, d) = p(\theta)$ in (11).

**Implicit likelihood and $\hat{\mu}_{\mathbf{m}+\ell}$**   So far we have assumed that we can evaluate $p(y|\theta, d)$ pointwise. However, many models of interest have *implicit likelihoods* from which we can draw samples, but not evaluate directly. For example, models with nuisance latent variables $\psi$ (such as a random effect models) are implicit likelihood models because $p(y|\theta, d) = \mathbb{E}_{p(\psi|\theta)}[p(y|\theta, \psi, d)]$ is intractable, but can still be straightforwardly sampled from.

In this setting, $\hat{\mu}_{\text{post}}$ is applicable without modification because it only requires samples from $p(y|\theta, d)$ and *not* evaluations of this density. Although $\hat{\mu}_{\text{marg}}$ is not directly applicable in this setting, it can be modified to accommodate implicit likelihoods. Specifically, we can utilize *two* approximate densities: $q_m(y|d)$ for the marginal and $q_\ell(y|\theta, d)$ for the likelihood. We then form the approximation

$$\text{EIG}(d) \approx \mathcal{I}_{\text{m}+\ell}(d) \triangleq \mathbb{E}_{p(y, \theta|d)}\left[\log \frac{q_\ell(y|\theta, d)}{q_m(y|d)}\right] \approx \hat{\mu}_{\text{m}+\ell}(d) \triangleq \frac{1}{N}\sum_{n=1}^{N} \log \frac{q_\ell(y_n|\theta_n, d)}{q_m(y_n|d)}. \quad (12)$$

Unlike the previous three cases, $\mathcal{I}_{\text{m}+\ell}(d)$ is not a bound on $\text{EIG}(d)$, meaning it is not immediately clear how to train $q_m(y|d)$ and $q_\ell(y|\theta, d)$ to achieve an accurate EIG estimator. The following lemma shows that we can bound the EIG estimation *error* of $\mathcal{I}_{\text{m}+\ell}$. The proof is in Appendix A.

**Lemma 2.** *For any given model $p(\theta)p(y|\theta, d)$ and valid $q_m(y|d)$ and $q_\ell(y|\theta, d)$, we have*

$$|\mathcal{I}_{m+\ell}(d) - \text{EIG}(d)| \leq -\mathbb{E}_{p(y, \theta|d)}[\log q_m(y|d) + \log q_\ell(y|\theta, d)] + C, \quad (13)$$

*where $C = -H[p(y|d)] - \mathbb{E}_{p(\theta)}[H(p(y|\theta, d))]$ does not depend on $q_m$ or $q_\ell$. Further, the RHS of (13) is 0 if and only if $q_m(y|d) = p(y|d)$ and $q_\ell(y|\theta, d) = p(y|\theta, d)$ for almost all $y, \theta$.*

This lemma implies that we can learn $q_m(y|d)$ and $q_\ell(y|\theta, d)$ by maximizing $\mathbb{E}_{p(y, \theta|d)}[\log q_m(y|d) + \log q_\ell(y|\theta, d)]$ using stochastic gradient ascent, and substituting these learned approximations into (12) for the final EIG estimator. To the best of our knowledge, this approach has not previously been considered in the literature. We note that, in general, $q_m$ and $q_\ell$ are learned separately and there need not be any weight sharing between them. See Appendix A.4 for a discussion of the case when we couple $q_m$ and $q_\ell$ so that $q_m(y|d) = \mathbb{E}_{p(\theta)}[q_\ell(y|\theta, d)]$.

**Using estimators for sequential BOED**   In sequential settings, we also need to consider the implications of replacing $p(\theta)$ in the EIG with $p(\theta|d_{1:t-1}, y_{1:t-1})$. At first sight, it appears that, while $\hat{\mu}_{\text{marg}}$ and $\hat{\mu}_{\text{m}+\ell}$ only require samples from $p(\theta|d_{1:t-1}, y_{1:t-1})$, $\hat{\mu}_{\text{post}}$ and $\hat{\mu}_{\text{VNMC}}$ also require its density to be evaluated, a potentially severe limitation. Fortunately, we can, in fact, avoid evaluating this posterior density. We note that, from (5), we have $p(\theta|y_{1:t-1}, d_{1:t-1}) = p(\theta)\prod_{i=1}^{t-1} p(y_i|\theta, d_i)/p(y_{1:t-1}|d_{1:t-1})$. Substituting this into the integrand of (6) gives

$$\mathcal{L}_{\text{post}}(d_t) = \mathbb{E}_{p(\theta|y_{1:t-1}, d_{1:t-1})p(y_t|\theta, d_t)}\left[\log \frac{q_p(\theta|y_t, d_t)}{p(\theta)\prod_{i=1}^{t-1} p(y_i|\theta, d_i)}\right] + \log p(y_{1:t-1}|d_{1:t-1}) \quad (14)$$

where $p(\theta)\prod_{i=1}^{t-1} p(y_i|\theta, d_i)$ can be evaluated exactly and the additive constant $\log p(y_{1:t-1}|d_{1:t-1})$ does not depend on the new design $d_t, \theta$, or any of the variational parameters, and so can be safely ignored. Making the same substitution in (11) shows that we can also estimate $\mathcal{U}_{\text{VNMC}}(d_t, L)$ up to a constant, which can then be similarly ignored. As such, any inference scheme for sampling $p(\theta|d_{1:t-1}, y_{1:t-1})$, approximate or exact, is compatible with all our approaches.

**Selecting an estimator**   Having proposed four estimators, we briefly discuss how to choose between them in practice. For reference, a summary of our estimators is given in Table 1, along with several baseline approaches. First, $\hat{\mu}_{\text{marg}}$ and $\hat{\mu}_{\text{m}+\ell}$ rely on approximating a distribution over $y$; $\hat{\mu}_{\text{post}}$ and $\hat{\mu}_{\text{VNMC}}$ approximate distributions over $\theta$. We may prefer the former two estimators if $\dim(y) \ll \dim(\theta)$ as it leaves us with a simpler density estimation problem, and vice versa. Second, $\hat{\mu}_{\text{marg}}$ and $\hat{\mu}_{\text{VNMC}}$ require an

Table 1: Summary of EIG estimators. Baseline methods are explained in Section 5.

|  |  | Implicit | Bound | Consistent | Eq. |
|---|---|---|---|---|---|
| Ours | $\hat{\mu}_{\text{post}}$ | ✓ | Lower | ✗ | (6) |
|  | $\hat{\mu}_{\text{marg}}$ | ✗ | Upper | ✗ | (9) |
|  | $\hat{\mu}_{\text{VNMC}}$ | ✗ | Upper | ✓ | (11) |
|  | $\hat{\mu}_{\text{m}+\ell}$ | ✓ | ✗ | ✗ | (12) |
| Baseline | $\hat{\mu}_{\text{NMC}}$ | ✗ | Upper | ✓ | (4) |
|  | $\hat{\mu}_{\text{laplace}}$ | ✗ | ✗ | ✗ | (75) |
|  | $\hat{\mu}_{\text{LFIRE}}$ | ✓ | ✗ | ✗ | (76) |
|  | $\hat{\mu}_{\text{DV}}$ | ✓ | Lower | ✗ | (77) |

explicit likelihood whereas $\hat{\mu}_{\text{post}}$ and $\hat{\mu}_{\text{m}+\ell}$ do not. If an explicit likelihood is available, it typically makes sense to use it—one would never use $\hat{\mu}_{\text{m}+\ell}$ over $\hat{\mu}_{\text{marg}}$ for example. Finally, if the variational families do not contain the target densities, $\hat{\mu}_{\text{VNMC}}$ is the only method guaranteed to converge to the true $\text{EIG}(d)$ in the limit as the computational budget increases. So we might prefer $\hat{\mu}_{\text{VNMC}}$ when computation time and cost are not constrained.

## 4   Convergence rates

We now investigate the convergence of our estimators. We start by breaking the overall error down into three terms: I) variance in MC estimation of the bound; II) the gap between the bound and the tightest bound possible given the variational family; and III) the gap between the tightest possible bound and $\text{EIG}(d)$. With variational EIG approximation $\mathcal{B}(d) \in \{\mathcal{L}_{\text{post}}(d), \mathcal{U}_{\text{marg}}(d), \mathcal{U}_{\text{VNMC}}(d, L), \mathcal{I}_{\text{m}+\ell}(d)\}$, optimal variational parameters $\phi^*$, learned variational parameters $\phi_K$ after $K$ stochastic gradient iterations, and MC estimator $\hat{\mu}(d, \phi_K)$ we have, by the triangle inequality,

$$\|\hat{\mu}(d, \phi_K) - \text{EIG}(d)\|_2 \leq \underbrace{\|\hat{\mu}(d, \phi_K) - \mathcal{B}(d, \phi_K)\|_2}_{\text{I}} + \underbrace{\|\mathcal{B}(d, \phi_K) - \mathcal{B}(d, \phi^*)\|_2}_{\text{II}} + \underbrace{|\mathcal{B}(d, \phi^*) - \text{EIG}(d)|}_{\text{III}}$$

where we have used the notation $\|X\|_2 \triangleq \sqrt{\mathbb{E}[X^2]}$ to denote the $L^2$ norm of a random variable.

By the weak law of large numbers, term I scales as $N^{-1/2}$ and can thus be arbitrarily reduced by taking more MC samples. Provided that our stochastic gradient scheme converges, term II can be reduced by increasing the number of stochastic gradient steps $K$. Term III, however, is a constant that can only be reduced by expanding the variational family (or increasing $L$ for $\hat{\mu}_{\text{VNMC}}$). Each approximation $\mathcal{B}(d)$ thus converges to a biased estimate of the $\text{EIG}(d)$, namely $\mathcal{B}(d, \phi^*)$. As established by the following Theorem, if we set $N \propto K$, the rate of convergence to this biased estimate is $\mathcal{O}(T^{-1/2})$, where $T$ represents the total computational cost, with $T = \mathcal{O}(N + K)$.

**Theorem 1.** *Let $\mathcal{X}$ be a measurable space and $\Phi$ be a convex subset of a finite dimensional inner product space. Let $X_1, X_2, \ldots$ be i.i.d. random variables taking values in $\mathcal{X}$ and $f : \mathcal{X} \times \Phi \to \mathbb{R}$ be a measurable function. Let*

$$\mu(\phi) \triangleq \mathbb{E}[f(X_1, \phi)] \approx \hat{\mu}_N(\phi) \triangleq \frac{1}{N} \sum_{n=1}^{N} f(X_n, \phi)$$

*and suppose that $\sup_{\phi \in \Phi} \|f(X_1, \phi)\|_2 < \infty$. Then $\sup_{\phi \in \Phi} \|\hat{\mu}_N(\phi) - \mu(\phi)\|_2 = \mathcal{O}(N^{-1/2})$. Suppose further that Assumption 1 in Appendix B holds and that $\phi^*$ is the unique minimizer of $\mu$. After $K$ iterations of the Polyak-Ruppert averaged stochastic gradient descent algorithm of [28] with gradient estimator $\nabla_\phi f(X_t, \phi)$, we have $\|\mu(\phi_K) - \mu(\phi^*)\|_2 = \mathcal{O}(K^{-1/2})$ and, combining with the first result,*

$$\|\hat{\mu}_N(\phi_K) - \mu(\phi^*)\|_2 = \mathcal{O}(N^{-1/2} + K^{-1/2}) = \mathcal{O}(T^{-1/2}) \text{ if } N \propto K.$$

The proof relies on standard results from MC and stochastic optimization theory; see Appendix B. We note that the assumptions required for the latter, though standard in the literature, are strong. In practice, $\phi$ can converge to a local optimum $\phi^\dagger$, rather than the global optimum $\phi^*$, introducing an additional asymptotic bias $|\mathcal{B}(d, \phi^\dagger) - \mathcal{B}(d, \phi^*)|$ into term III.

Theorem 1 can be applied directly to $\hat{\mu}_{\text{marg}}$, $-\hat{\mu}_{\text{post}}$, and $\hat{\mu}_{\text{VNMC}}$ (with fixed $M = L$), showing that they converge respectively to $\mathcal{U}_{\text{marg}}(d, \phi^*)$, $-\mathcal{L}_{\text{post}}(d, \phi^*)$, and $\mathcal{U}_{\text{VNMC}}(d, L, \phi^*)$ at a rate $= \mathcal{O}(T^{-1/2})$ if $N \propto K$ and the assumptions are satisfied. For $\hat{\mu}_{\text{m}+\ell}$, we combine Theorem 1 and Lemma 2 to obtain the same $\mathcal{O}(T^{-1/2})$ convergence rates; see the supplementary material for further details.

The key property of $\hat{\mu}_{\text{VNMC}}$ is that we need not set $M = L$ and can remove the asymptotic bias by increasing $M$ with $N$. We begin by training $\phi$ with a fixed value of $L$, decreasing the error term $\|\mathcal{U}_{\text{VNMC}}(d, L, \phi_K) - \mathcal{U}_{\text{VNMC}}(d, L, \phi^*)\|_2$ at the fast rate $\mathcal{O}(K^{-1/2})$ until $|\mathcal{U}_{\text{VNMC}}(d, L, \phi^*) - \text{EIG}(d)|$ becomes the dominant error term. At this point, we start to increase $N, M$. Using the NMC convergence results discussed in Sec. 2, if we set $M \propto \sqrt{N}$, then $\hat{\mu}_{\text{VNMC}}$ converges to $\text{EIG}(d)$ at a rate $\mathcal{O}((NM)^{-1/3})$. Note that the total cost of the $\hat{\mu}_{\text{VNMC}}$ estimator is $T = \mathcal{O}(KL + NM)$, where typically $M \gg L$. The first stage, costing $KL$, is fast variational training of an amortized importance sampling proposal for $p(y|d) = \mathbb{E}_{p(\theta)}[p(y|\theta, d)]$. The second stage, costing $NM$, is slower refinement to remove the asymptotic bias using the learned proposal in an NMC estimator.

Table 2: Bias squared and variance from 5 runs, averaged over designs, of EIG estimators applied to four benchmarks. We use - to denote that a method does not apply and $*$ when it is superseded by other methods. Bold indicates the estimator with the lowest empirical mean squared error.

| | A/B test | | Preference | | Mixed effects | | Extrapolation | |
|---|---|---|---|---|---|---|---|---|
| | Bias$^2$ | Var | Bias$^2$ | Var | Bias$^2$ | Var | Bias$^2$ | Var |
| $\hat{\mu}_{\text{post}}$ | $1.33\times10^{-2}$ | $7.15\times10^{-3}$ | $4.26\times10^{-2}$ | $8.53\times10^{-3}$ | $2.34\times10^{-3}$ | $2.92\times10^{-3}$ | $1.24\times10^{-4}$ | $5.16\times10^{-5}$ |
| $\hat{\mu}_{\text{marg}}$ | $7.45\times10^{-2}$ | $6.41\times10^{-3}$ | $\mathbf{1.10\times10^{-3}}$ | $\mathbf{1.99\times10^{-3}}$ | - | - | - | - |
| $\hat{\mu}_{\text{VNMC}}$ | $3.44\times10^{-3}$ | $3.38\times10^{-3}$ | $4.17\times10^{-3}$ | $9.04\times10^{-3}$ | - | - | - | - |
| $\hat{\mu}_{\text{m}+\ell}$ | $*$ | $*$ | $*$ | $*$ | $\mathbf{3.06\times10^{-3}}$ | $\mathbf{5.94\times10^{-5}}$ | $\mathbf{6.90\times10^{-6}}$ | $\mathbf{1.84\times10^{-5}}$ |
| $\hat{\mu}_{\text{NMC}}$ | $4.70\times10^{0}$ | $3.47\times10^{-1}$ | $7.60\times10^{-2}$ | $8.36\times10^{-2}$ | - | - | - | - |
| $\hat{\mu}_{\text{laplace}}$ | $\mathbf{1.92\times10^{-4}}$ | $\mathbf{1.47\times10^{-3}}$ | $8.42\times10^{-2}$ | $9.70\times10^{-2}$ | - | - | - | - |
| $\hat{\mu}_{\text{LFIRE}}$ | $2.29\times10^{0}$ | $6.20\times10^{-1}$ | $1.30\times10^{-1}$ | $1.41\times10^{-2}$ | $1.41\times10^{-1}$ | $6.67\times10^{-2}$ | - | - |
| $\hat{\mu}_{\text{DV}}$ | $4.34\times10^{0}$ | $8.85\times10^{-1}$ | $9.23\times10^{-2}$ | $8.07\times10^{-3}$ | $9.10\times10^{-3}$ | $5.56\times10^{-4}$ | $7.84\times10^{-6}$ | $4.11\times10^{-5}$ |

One can think of the standard NMC approach as a special case of $\hat{\mu}_{\text{VNMC}}$ in which we naively choose $p(\theta)$ as the proposal. That is, standard NMC skips the first stage and hence does not benefit from the improved convergence rate of learning an amortized proposal. It typically requires a much higher total cost to achieve the same accuracy as VNMC.

## 5   Related work

We briefly discuss alternative approaches to EIG estimation for BOED that will form our baselines for empirical comparisons. The **Nested Monte Carlo (NMC)** baseline was introduced in Sec. 2. Another established approach is to use a **Laplace approximation** to the posterior [22, 25]; this approach is fast but is limited to continuous variables and can exhibit large bias. Kleinegesse and Gutmann [18] recently suggested an implicit likelihood approach based on the Likelihood-Free Inference by Ratio Estimation **(LFIRE)** method of Thomas et al. [41]. We also consider a method based on the **Donsker-Varadhan (DV)** representation of the KL divergence [11] as used by Belghazi et al. [4] for mutual information estimation. Though not previously considered in BOED, we include it as a baseline for illustrative purposes. For a full discussion of the DV bound and a number of other variational bounds used in deep learning, we refer to the recent work of Poole et al. [31]. For further discussion of related work, see Appendix C.

## 6   Experiments

### 6.1   EIG estimation accuracy

We begin by benchmarking our EIG estimators against the aforementioned baselines. We consider four experiment design scenarios inspired by applications of Bayesian data analysis in science and industry. First, **A/B testing** is used across marketing and design [6, 19] to study population traits. Here, the design is the choice of the A and B group sizes and the Bayesian model is a Gaussian linear model. Second, revealed **preference** [36] is used in economics to understand consumer behaviour. We consider an experiment design setting in which we aim to learn the underlying utility function of an economic agent by presenting them with a proposal (such as offering them a price for a commodity) and observing their revealed preference. Third, fixed effects and random effects (nuisance variables) are combined in **mixed effects** models [14, 20]. We consider an example inspired by item-response theory [13] in psychology. We seek information only about the fixed effects, making this an implicit likelihood problem. Finally, we consider an experiment where labelled data from one region of design space must be used to predict labels in a target region by **extrapolation** [27]. In summary, we have two models with explicit likelihoods (A/B testing, preference) and two that are implicit (mixed effects, extrapolation). Full details of each model are presented in Appendix D.

For each scenario, we estimated the EIG across a grid of designs with a fixed computational budget for each estimator and calculated the true EIG analytically or with brute force computation as appropriate; see Table 2 for the results. Whilst the Laplace method, unsurprisingly, performed best for the Gaussian linear model where its approximation becomes exact, we see that our methods are otherwise more accurate. All our methods outperformed NMC.

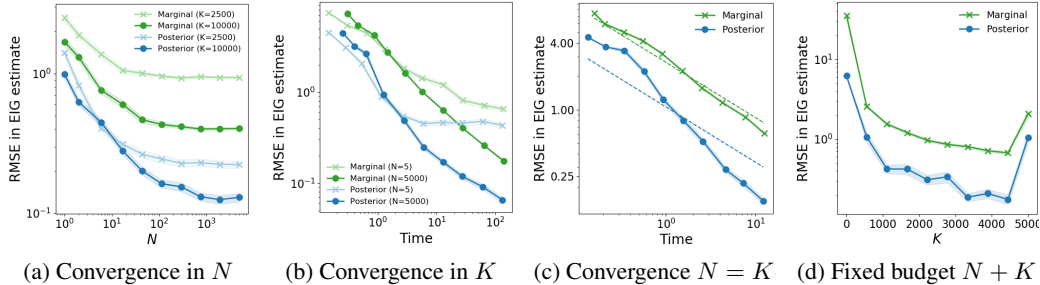

|(a) Convergence in $N$ | (b) Convergence in $K$ | (c) Convergence $N = K$ | (d) Fixed budget $N + K$|

Figure 1: Convergence of RMSE for $\hat{\mu}_{\text{post}}$ and $\hat{\mu}_{\text{marg}}$. (a) Convergence in number of MC samples $N$ with a fixed number $K$ of gradient updates of the variational parameters. (b) Convergence in time when increasing $K$ and with $N$ fixed. (c) Convergence in time when setting $N = K$ and increasing both (dashed lines represent theoretical rates). (d) Final RMSE with $N + K = 5000$ fixed, for different $K$. Each graph shows the mean with shading representing $\pm 1$ std. err. from 100 trials.

## 6.2 Convergence rates

We now investigate the empirical convergence characteristics of our estimators. Throughout, we consider a single design point from the A/B test example. We start by examining the convergence of $\hat{\mu}_{\text{post}}$ and $\hat{\mu}_{\text{marg}}$ as we allocate the computational budget in different ways.

We first consider the convergence in $N$ after a fixed number of $K$ updates to the variational parameters. As shown in Figure 1a, the RMSE initially decreases as we increase $N$, before plateauing due to the bias in the estimator. We also see that $\hat{\mu}_{\text{post}}$ substantially outperforms $\hat{\mu}_{\text{marg}}$. We next consider the convergence as a function of wall-clock time when $N$ is held fixed and we increase $K$. We see in Figure 1b that, as expected, the errors decrease with time and that when a small value of $N = 5$ is taken, we again see a plateauing effect, with the variance of the final MC estimator now becoming the limiting factor. In Figure 1c we take $N = K$ and increase both, obtaining the predicted convergence rate $\mathcal{O}(T^{-1/2})$ (shown by the dashed lines). We conjecture that the better performance of $\hat{\mu}_{\text{post}}$ is likely due to $\theta$ being lower dimensional ($\dim = 2$) than $y$ ($\dim = 10$). In Figure 1d, we instead fix $T = N + K$ to investigate the optimal trade-off between optimization and MC error: it appears the range of $K/T$ between $0.5$ and $0.9$ gives the lowest RMSE.

Finally, we show how $\hat{\mu}_{\text{VNMC}}$ can improve over NMC by using an improved variational proposal for estimating $p(y|d)$. In Figure 2, we plot the EIG estimates obtained by first running $K$ steps of stochastic gradient with $L = 1$ to learn $q_v(\theta|y,d)$, before increasing $M$ and $N$. We see that spending some of our time budget training $q_v(\theta|y,d)$ leads to noticeable improvements in the estimation, but also that it is important to increase $N$ and $M$. Rather than plateauing like $\hat{\mu}_{\text{post}}$ and $\hat{\mu}_{\text{marg}}$, $\hat{\mu}_{\text{VNMC}}$ continues to improve after the initial training period as, albeit at a slower $O(T^{-1/3})$ rate.

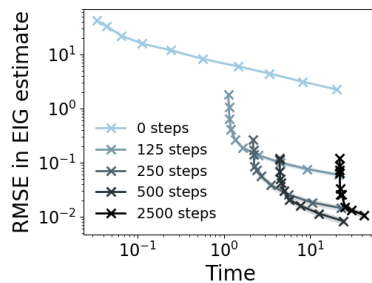

Figure 2: Convergence of $\hat{\mu}_{\text{VNMC}}$ taking $M = \sqrt{N}$. 'Steps' refers to pre-training of the variational posterior (i.e. $K$), with 0 steps corresponding to $\hat{\mu}_{\text{NMC}}$. Means and confidence intervals as per Fig. 1.

## 6.3 End-to-end sequential experiments

We now demonstrate the utility of our methods for designing sequential experiments. First, we demonstrate that our variational estimators are sufficiently robust and fast to be used for adaptive experiments with a class of models that are of practical importance in many scientific disciplines. To this end, we run an adaptive psychology experiment with human participants recruited from Amazon Mechanical Turk to study how humans respond to features of stylized faces. To account for fixed effects—those *common* across the population—as well as individual variations that we treat as nuisance variables, we use the mixed effects regression model introduced in Sec. 6.1. See Appendix D for full details of the experiment.

To estimate the EIG for different designs, we use $\hat{\mu}_{\text{m}+\ell}$, since it yields the best performance on our mixed effects model benchmark (see Table 2). Our EIG estimator is integrated into a system that

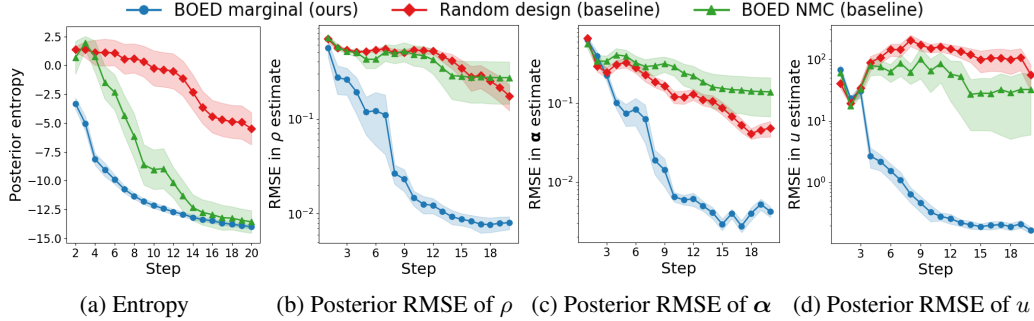

(a) Entropy     (b) Posterior RMSE of $\rho$    (c) Posterior RMSE of $\boldsymbol{\alpha}$    (d) Posterior RMSE of $u$

Figure 4: Evolution of the posterior in the sequential CES experiment. (a) Total entropy of a mean-field variational approximation of the posterior. (b)(c)(d) The RMSE of the posterior approximations of $\rho$, $\boldsymbol{\alpha}$ and $u$ as compared to the true values used to simulate agent responses. Note the scale of the vertical axis is logarithmic. All plots show the mean and $\pm 1$ std. err. from 10 independent runs.

presents participants with a stimulus, receives their response, learns an updated model, and designs the next stimulus, all online. Despite the relative simplicity of the design problem (with 36 possible designs) using BOED with $\hat{\mu}_{m+\ell}$ leads to a more certain (i.e. lower entropy) posterior than random design; see Figure 3.

Second, we consider a more challenging scenario in which a random design strategy gleans very little. We compare random design against two BOED strategies: $\hat{\mu}_{\mathrm{marg}}$ and $\hat{\mu}_{\mathrm{NMC}}$. Building on the revealed preference example in Sec. 6.1, we consider an experiment to infer an agent's utility function which we model using the Constant Elasticity of Substitution (CES) model [2] with latent variables $\rho, \boldsymbol{\alpha}, u$. We seek designs for which the agent's response will be informative about $\theta = (\rho, \boldsymbol{\alpha}, u)$. See Appendix D for full details. We estimate the EIG using $\hat{\mu}_{\mathrm{marg}}$ because the dimension of $y$ is smaller than that of $\theta$, and select designs $d \in [0, 100]^6$ using Bayesian optimization. To investigate parameter recovery we simulate agent responses from the model with fixed values of $\rho, \boldsymbol{\alpha}, u$. Figure 4 shows that using BOED with our marginal estimator reduces posterior entropy *and* concentrates

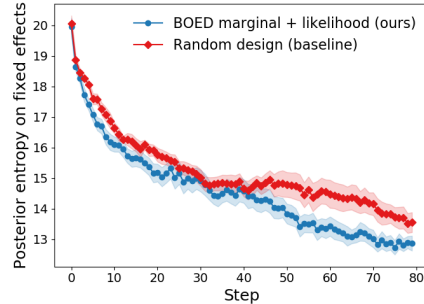

Figure 3: Evolution of the posterior entropy of the fixed effects in the Mechanical Turk experiment in Sec. 6.3. We depict the mean and $\pm 1$ std. err. from 10 experimental trials.

more quickly on the true parameter values than both baselines. Random design makes no inroads into the learning problem, while BOED based on NMC particularly struggles at the outset when $p(\theta|d_{1:t-1}, y_{1:t-1})$, the prior at iteration $t$, is high variance. Our method selects informative designs throughout.

# 7   Discussion

We have developed efficient EIG estimators that are applicable to a wide range of experimental design problems. By tackling the double intractability of the EIG in a principled manner, they provide substantially improved convergence rates relative to previous approaches, and our experiments show that these theoretical advantages translate into significant practical gains. Our estimators are well-suited to modern deep probabilistic programming languages and we have provided an implementation in Pyro. We note that the interplay between variational and MC methods in EIG estimation is not directly analogous to those in standard inference settings because the NMC EIG estimator is itself inherently biased. Our $\hat{\mu}_{\mathrm{VNMC}}$ estimator allows one to play off the advantages of these approaches, namely the fast learning of variational approaches and asymptotic consistency of NMC.

## Acknowledgements

We gratefully acknowledge research funding from Uber AI Labs. MJ would like to thank Paul Szerlip for help generating the sprites used in the Mechanical Turk experiment. AF would like to thank Patrick Rebeschini, Dominic Richards and Emile Mathieu for their help and support. AF gratefully acknowledges funding from EPSRC grant no. EP/N509711/1. YWT's and TR's research leading to these results has received funding from the European Research Council under the European Union's Seventh Framework Programme (FP7/2007-2013) ERC grant agreement no. 617071.

## Footnotes

* Part of this work was completed by AF during an internship with Uber AI Labs.

[2]Implementations of our methods are available at `http://docs.pyro.ai/en/stable/contrib.oed.html`. To reproduce the results in this paper, see `https://github.com/ae-foster/pyro/tree/vboed-reproduce`.

[3]See Appendix A for a proof. A comparison with the reverse KL divergence can be found in Appendix G.

[4]In Appendix F, we describe a method using $q_m(y|d)$ as a control variate that can also eliminate this bias and lower the variance of NMC, requiring additional assumptions about the model and variational family.

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
