[Supplementary Material · variational_boed_supplement.pdf]

# Supplement for Variational Bayesian Optimal Experimental Design

**Adam Foster**[†]   **Martin Jankowiak**[‡]   **Eli Bingham**[‡]   **Paul Horsfall**[‡]
**Yee Whye Teh**[†]   **Tom Rainforth**[†]   **Noah Goodman**[‡§]
[†]Department of Statistics, University of Oxford, Oxford, UK
[‡]Uber AI Labs, Uber Technologies Inc., San Francisco, CA, USA
[§]Stanford University, Stanford, CA, USA
adam.foster@stats.ox.ac.uk

## A   Details for variational estimators

The proofs in A.1 and A.2 are included for completeness.

### A.1   Variational posterior $\hat{\mu}_{\text{post}}$

We require valid approximations $q_p(\theta|y,d)$ to have the same support as $p(\theta|y,d)$. Recall

$$\mathcal{L}_{\text{post}}(d) = \mathbb{E}_{p(y,\theta|d)}\left[\log \frac{q_p(\theta|y,d)}{p(\theta)}\right] \tag{16}$$

and

$$\text{EIG}(d) = \mathbb{E}_{p(y,\theta|d)}\left[\log \frac{p(\theta|y,d)}{p(\theta)}\right] \tag{17}$$

We aim to show $\text{EIG}(d) \geq \mathcal{L}_{\text{post}}(d)$. Following [3], we have

$$\text{EIG}(d) - \mathcal{L}_{\text{post}}(d) = \mathbb{E}_{p(y,\theta|d)}\left[\log \frac{p(\theta|y,d)}{p(\theta)} - \log \frac{q_p(\theta|y,d)}{p(\theta)}\right] \tag{18}$$

$$= \mathbb{E}_{p(y,\theta|d)}\left[\log \frac{p(\theta|y,d)p(\theta)}{p(\theta)q_p(\theta|y,d)}\right] \tag{19}$$

$$= \mathbb{E}_{p(y|d)}\left[\mathbb{E}_{p(\theta|y,d)}\left[\log \frac{p(\theta|y,d)}{q_p(\theta|y,d)}\right]\right] \tag{20}$$

$$= \mathbb{E}_{p(y|d)}\left[\text{KL}\left(p(\theta|y,d)||q_p(\theta|y,d)\right)\right] \tag{21}$$

$$\geq 0. \tag{22}$$

To further prove that the bound is tight, we note that the penultimate term $\mathbb{E}_{p(y|d)}\left[\text{KL}\left(p(\theta|y,d)||q_p(\theta|y,d)\right)\right]$ equals 0 if and only if $\text{KL}\left(p(\theta|y,d)||q_p(\theta|y,d)\right) = 0$ for almost all $y$ (i.e. the union of all $y$ for which this does not hold has measure zero). The occurs if and only if $q_p(\theta|y,d) = p(\theta|y,d)$ for almost all $y, \theta$.

### A.2   Variational marginal $\hat{\mu}_{\text{marg}}$

We now demonstrate that $\mathcal{U}_{\text{marg}}(d)$ is an upper bound on $\text{EIG}(d)$. Proceeding in the same manner as for $\hat{\mu}_{\text{post}}$, we find

$$\mathcal{U}_{\text{marg}}(d) - \text{EIG}(d) = \mathbb{E}_{p(y,\theta|d)}\left[\log \frac{p(y|\theta,d)}{q_m(y|d)} - \log \frac{p(y|\theta,d)}{p(y|d)}\right] \tag{23}$$

$$=\mathbb{E}_{p(y,\theta|d)}\left[\log\frac{p(y|\theta,d)p(y|d)}{q_m(y|d)p(y|\theta,d)}\right] \tag{24}$$

$$=\mathbb{E}_{p(y|d)}\left[\log\frac{p(y|d)}{q_m(y|d)}\right] \tag{25}$$

$$=\mathrm{KL}\left(p(y|d)||q_m(y|d)\right) \tag{26}$$

$$\geq 0. \tag{27}$$

Again, the bound is tight if and only if $q_m(y|d) = p(y|d)$ almost everywhere.

### A.3 Variational NMC $\hat{\mu}_{\textbf{VNMC}}$

We now prove Lemma 1 from the main paper, duplicating the Lemma itself below for convenience.

**Lemma 1.** *For any given model $p(\theta)p(y|\theta, d)$ and valid $q_v(\theta|y, d)$,*

1. $\mathrm{EIG}(d) = \lim_{L\to\infty}\mathcal{U}_{VNMC}(d, L) \leq \mathcal{U}_{VNMC}(d, L_2) \leq \mathcal{U}_{VNMC}(d, L_1) \quad \forall L_2 \geq L_1 \geq 1,$

2. $\mathcal{U}_{VNMC}(d, L) = \mathrm{EIG}(d) \ \ \forall L \geq 1 \ \ if \ \ q_v(\theta|y, d) = p(\theta|y, d) \ \ \forall y, \theta,$

3. $\mathcal{U}_{VNMC}(d, L) - \mathrm{EIG}(d) = \mathbb{E}_{p(y|d)}\left[\mathrm{KL}\left(\prod_{\ell=1}^{L} q_v(\theta_\ell|y, d)||\frac{1}{L}\sum_{\ell=1}^{L} p(\theta_\ell|y, d)\prod_{k\neq\ell} q_v(\theta_k|y, d)\right)\right]$

*Proof.* Starting with proving the first result in lemma, we first recall the definition of $\mathcal{U}_{\mathrm{VNMC}}(d, L)$ itself,

$$\mathcal{U}_{\mathrm{VNMC}}(d, L) = \mathbb{E}\left[\log p(y|\theta_0, d) - \log\frac{1}{L}\sum_{\ell=1}^{L}\frac{p(y, \theta_\ell|d)}{q_v(\theta_\ell|y, d)}\right] \tag{28}$$

where the expectation is taken over $y, \theta_{0:L} \sim p(y, \theta_0|d)\prod_{\ell=1}^{L} q_v(\theta_\ell|y, d)$. We consider positive integers $L_2 \geq L_1$. We let $\delta = \mathcal{U}_{\mathrm{VNMC}}(d, L_1) - \mathcal{U}_{\mathrm{VNMC}}(d, L_2)$. Then,

$$\delta = \mathbb{E}\left[\log\frac{1}{L_2}\sum_{\ell=1}^{L_2}\frac{p(y, \theta_\ell|d)}{q_v(\theta_\ell|y, d)}\right] - \mathbb{E}\left[\log\frac{1}{L_1}\sum_{\ell=1}^{L_1}\frac{p(y, \theta_\ell|d)}{q_v(\theta_\ell|y, d)}\right]. \tag{29}$$

We now proceed as in [7]. Let $I_1, ..., I_{L_1}$ be distinct indices drawn uniformly from $1, ..., L_2$. Then,

$$\frac{1}{L_2}\sum_{\ell=1}^{L_2}\frac{p(y, \theta_\ell)}{q_v(\theta_\ell|y, d)} = \mathbb{E}_{I_1, ..., I_{L_1}}\left[\frac{1}{L_1}\sum_{j=1}^{L_1}\frac{p(y, \theta_{I_j})}{q_v(\theta_{I_j}|y, d)}\right] \tag{30}$$

So

$$\delta = \mathbb{E}\left[\log\left(\mathbb{E}_{I_{1:L_1}}\left[\frac{1}{L_1}\sum_{j=1}^{L_1}\frac{p(y, \theta_{I_j})}{q_v(\theta_{I_j}|y, d)}\right]\right)\right] - \mathbb{E}\left[\log\frac{1}{L_1}\sum_{\ell=1}^{L_1}\frac{p(y, \theta_\ell|d)}{q_v(\theta_\ell|y, d)}\right], \tag{31}$$

then by Jensen's Inequality

$$\delta \geq \mathbb{E}\left[\mathbb{E}_{I_{1:L_1}}\left[\log\left(\frac{1}{L_1}\sum_{j=1}^{L_1}\frac{p(y, \theta_{I_j})}{q_v(\theta_{I_j}|y, d)}\right)\right]\right] - \mathbb{E}\left[\log\frac{1}{L_1}\sum_{\ell=1}^{L_1}\frac{p(y, \theta_\ell|d)}{q_v(\theta_\ell|y, d)}\right] \tag{32}$$

$$\geq \mathbb{E}\left[\log\frac{1}{L_1}\sum_{\ell=1}^{L_1}\frac{p(y, \theta_\ell|d)}{q_v(\theta_\ell|y, d)}\right] - \mathbb{E}\left[\log\frac{1}{L_1}\sum_{\ell=1}^{L_1}\frac{p(y, \theta_\ell|d)}{q_v(\theta_\ell|y, d)}\right] \tag{33}$$

$$\geq 0 \tag{34}$$

where we have used that $\theta_{I_1}, ..., \theta_{I_{L_1}} \overset{d}{=} \theta_1, ..., \theta_{L_1}$. This shows that $\mathcal{U}_{\mathrm{VNMC}}(d, L_1) \geq \mathcal{U}_{\mathrm{VNMC}}(d, L_2)$. For the limit $\lim_{L\to\infty}\mathcal{U}_{\mathrm{VNMC}}(d, L)$ we first fix some $y$ for which $p(y|d) > 0$ and consider

$$\mathcal{U}_{\mathrm{VNMC}}(d, L, y) = \mathbb{E}\left[\log p(y|\theta_0, d) - \log\frac{1}{L}\sum_{\ell=1}^{L}\frac{p(y, \theta_\ell|d)}{q_v(\theta_\ell|y, d)}\right]. \tag{35}$$

with the expectation taken over $p(\theta_0|y,d) \prod_{\ell=1}^{L} q_v(\theta_\ell|y,d)$. Since $p(y,\theta|d)/q_v(\theta|y,d)$ is bounded by assumption, the Strong Law of Large Numbers implies that, in limit of large $L$,

$$\frac{1}{L} \sum_{\ell=1}^{L} \frac{p(y,\theta_\ell|d)}{q_v(\theta_\ell|y,d)} \to p(y|d) \ a.s. \tag{36}$$

Furthermore, using the same argument as before, $\mathcal{U}_{\text{VNMC}}(d,L_1,y) \geq \mathcal{U}_{\text{VNMC}}(d,L_2,y)$ whenever $L_2 \geq L_1$. Thus the Bounded Convergence Theorem implies

$$\mathcal{U}_{\text{VNMC}}(d,L,y) \downarrow \mathbb{E}_{p(\theta_0|y,d)}[\log p(y|\theta_0,d) - \log p(y|d)] \text{ as } L \to \infty \tag{37}$$

so, taking expectations of $p(y|d)$, by the Monotone Convergence Theorem

$$\mathcal{U}_{\text{VNMC}}(d,L) \downarrow \mathbb{E}_{p(y,\theta_0|d)}[\log p(y|\theta_0,d) - \log p(y|d)] = \text{EIG}(d) \text{ as } L \to \infty. \tag{38}$$

For the second result, we simply note that

$$\frac{p(y,\theta|d)}{p(\theta|y,d)} = \frac{p(y,\theta|d)}{\frac{p(y,\theta|d)}{p(y|d)}} = p(y|d) \tag{39}$$

Finally, for the third result, we proceed as in [21]. We have

$$\mathcal{U}_{\text{VNMC}}(d,L) - \text{EIG}(d) = \mathbb{E}\left[\log p(y|d) - \log \frac{1}{L}\sum_{\ell=1}^{l} \frac{p(y,\theta_\ell|d)}{q_v(\theta_\ell|y,d)}\right] \tag{40}$$

where the expectation is over $p(y,\theta_0|d)\prod_{\ell=1}^{L} q_v(\theta_\ell|y,d)$.

Then

$$\mathcal{U}_{\text{VNMC}}(d,L) - \text{EIG}(d) = \mathbb{E}\left[-\log \frac{1}{L}\sum_{\ell=1}^{L} \frac{p(\theta_\ell|y,d)}{q_v(\theta_\ell|y,d)}\right] \tag{41}$$

$$= \mathbb{E}\left[\log \frac{\prod_{\ell=1}^{L} q_v(\theta_\ell|y,d)}{\frac{1}{L}\sum_{\ell=1}^{L} p(\theta_\ell|y,d)\prod_{k\neq\ell} q_v(\theta_k|y,d)}\right] \tag{42}$$

$$= \mathbb{E}\left[\log \frac{\prod_{\ell=1}^{L} q_v(\theta_\ell|y,d)}{P(\theta_{1:L}|y,d)}\right] \tag{43}$$

$$= \mathbb{E}_{p(y|d)}\left[\text{KL}\left(\prod_{\ell=1}^{L} q_v(\theta_\ell|y,d)||P(\theta_{1:L}|y,d)\right)\right] \tag{44}$$

where $P(\theta_{1:L}|y,d) = \frac{1}{L}\sum_{\ell=1}^{L} p(\theta_\ell|y,d)\prod_{k\neq\ell} q_v(\theta_k|y,d)$. □

### A.4 Variational marginal + likelihood $\hat{\mu}_{\mathbf{m}+\ell}$

We now prove Lemma 2 from the main paper, duplicating the Lemma itself below for convenience.

**Lemma 2.** *For any given model $p(\theta)p(y|\theta,d)$ and valid $q_m(y|d)$ and $q_\ell(y|\theta,d)$, we have*

$$|\mathcal{I}_{m+\ell}(d) - \text{EIG}(d)| \leq -\mathbb{E}_{p(y,\theta|d)}[\log q_m(y|d) + \log q_\ell(y|\theta,d)] + C, \tag{13}$$

*where $C = -H[p(y|d)] - \mathbb{E}_{p(\theta)}[H(p(y|\theta,d))]$ does not depend on $q_m$ or $q_\ell$. Further, the RHS of (13) is 0 if and only if $q_m(y|d) = p(y|d)$ and $q_\ell(y|\theta,d) = p(y|\theta,d)$ for almost all $y,\theta$.*

*Proof.* We aim to bound $|\mathcal{I}_{m+\ell}(d) - \text{EIG}(d)|$. Let $\delta = \mathcal{I}_{m+\ell}(d) - \text{EIG}(d)$. We have

$$\delta = \mathbb{E}_{p(y,\theta|d)}\left[\log \frac{q_\ell(y|\theta,d)}{q_m(y|d)}\right] - \mathbb{E}_{p(y,\theta|d)}\left[\log \frac{p(y|\theta,d)}{p(y|d)}\right] \tag{45}$$

$$= \mathbb{E}_{p(y,\theta|d)}\left[\log \frac{q_\ell(y|\theta,d)}{q_m(y|d)} - \log \frac{p(y|\theta,d)}{p(y|d)}\right] \tag{46}$$

$$= \mathbb{E}_{p(y,\theta|d)} \left[ \log \frac{q_\ell(y|\theta,d)}{q_m(y|d)} - \log \frac{p(y|\theta,d)}{q_m(y|d)} + \log \frac{p(y|\theta,d)}{q_m(y|d)} - \log \frac{p(y|\theta,d)}{p(y|d)} \right] \tag{47}$$

$$= -\mathbb{E}_{p(y,\theta|d)} \left[ \log \frac{q_m(y|d)p(y|\theta,d)}{q_\ell(y|\theta,d)q_m(y|d)} \right] + \mathbb{E}_{p(y,\theta|d)} \left[ \log \frac{p(y|\theta,d)p(y|d)}{q_m(y|d)p(y|\theta,d)} \right] \tag{48}$$

$$= -\mathbb{E}_{p(\theta)} \left[ \mathbb{E}_{p(y|\theta,d)} \left[ \log \frac{p(y|\theta,d)}{q_\ell(y|\theta,d)} \right] \right] + \mathbb{E}_{p(y|d)} \left[ \log \frac{p(y|d)}{q_m(y|d)} \right] \tag{49}$$

$$= -\mathbb{E}_{p(\theta)} \left[ \mathrm{KL}(p(y|\theta,d)||q_\ell(y|\theta,d)) \right] + \mathrm{KL}(p(y|d)||q_m(y|d)). \tag{50}$$

So, by the triangle inequality

$$|\delta| \le \mathbb{E}_{p(\theta)} \left[ \mathrm{KL}(p(y|\theta,d)||q_\ell(y|\theta,d)) \right] + \mathrm{KL}(p(y|d)||q_m(y|d)). \tag{51}$$

We can rewrite the RHS using the following relation

$$\mathrm{KL}(p(x)||q(x)) = \mathbb{E}_{p(x)} \left[ \log \frac{p(x)}{q(x)} \right] \tag{52}$$

$$= \mathbb{E}_{p(x)}[\log p(x)] - \mathbb{E}_{p(x)}[\log q(x)] \tag{53}$$

$$= -H[p(x)] - \mathbb{E}_{p(x)}[\log q(x)]. \tag{54}$$

This gives us

$$|\delta| \le \mathbb{E}_{p(\theta)} \left[ -H(p(y|\theta,d)] - \mathbb{E}_{p(y,\theta|d)}[\log q_\ell(y|\theta,d)] - H[p(y|d)] - \mathbb{E}_{p(y,|d)}[\log q_m(y|d)] \right. \tag{55}$$

$$\le -\mathbb{E}_{p(y,\theta|d)}[\log q_m(y|d) + \log q_\ell(y|\theta,d)] - H[p(y|d)] - \mathbb{E}_{p(\theta)} \left[ H(p(y|\theta,d)] \right] \tag{56}$$

as required.

Finally, from (51) we see that the error bound is tight if and only if both KL-divergences are 0 if and only if $q_\ell(y|\theta,d) = p(y|\theta,d)$ and $q_m(y|d) = p(y|d)$ for almost all $y, \theta$. $\qquad\square$

We conclude with an additional observation. Suppose that we set $q_m(y|d) = \mathbb{E}_{p(\theta)}[q_\ell(y|\theta,d)]$. This could be possible for instance when $\theta$ takes finitely many values. In this case, $\mathcal{I}_{\mathrm{m}+\ell}(d)$ is actually a lower bound on $\mathrm{EIG}(d)$. This is in contrast to the general case when $q_m$ and $q_\ell$ are learned separately, in which it is neither an upper nor a lower bound.

To show that $\mathcal{I}_{\mathrm{m}+\ell}(d)$ is a lower bound when $q_m(y|d) = \mathbb{E}_{p(\theta)}[q_\ell(y|\theta,d)]$, we begin with the Donsker-Varadhan bound [11]

$$\mathrm{EIG}(d) \ge \mathbb{E}_{p(y,\theta|d)}[T(y,\theta)] - \log \left( \mathbb{E}_{p(\theta)p(y|d)}[e^{T(y,\theta)}] \right). \tag{57}$$

Substituting $T(y,\theta) = \log(q_\ell(y|\theta,d)/q_m(y|d))$ we have

$$\mathrm{EIG}(d) \ge \mathbb{E}_{p(y,\theta|d)} \left[ \log \frac{q_\ell(y|\theta,d)}{q_m(y|d)} \right] - \log \left( \mathbb{E}_{p(\theta)p(y|d)} \left[ \frac{q_\ell(y|\theta,d)}{q_m(y|d)} \right] \right) \tag{58}$$

$$\ge \mathcal{I}_{\mathrm{m}+\ell}(d) - \log \left( \mathbb{E}_{p(y|d)} \left[ \mathbb{E}_{p(\theta)} \left\{ \frac{q_\ell(y|\theta,d)}{q_m(y|d)} \right\} \right] \right) \tag{59}$$

$$\ge \mathcal{I}_{\mathrm{m}+\ell}(d) - \log \left( \mathbb{E}_{p(y|d)} \left[ \frac{\mathbb{E}_{p(\theta)} \left\{ q_\ell(y|\theta,d) \right\}}{q_m(y|d)} \right] \right) \tag{60}$$

$$\ge \mathcal{I}_{\mathrm{m}+\ell}(d) - \log \left( \mathbb{E}_{p(y|d)} \left[ \frac{q_m(y|d)}{q_m(y|d)} \right] \right) \tag{61}$$

$$\ge \mathcal{I}_{\mathrm{m}+\ell}(d). \tag{62}$$

## B  Details for convergence rates

We now provide the details for Theorem 1. Key to proving the aspect of the Theorem relating to the convergence of the variational parameter $\phi_K$ to $\phi^*$ is Assumption 1. Points 1-5 correspond to assumptions H2', H3, H4, H6, and H7 of [28]; our proof will rely heavily on theirs. We note that also that our measurability assumption made in the Theorem itself means that their assumption H1 is automatically satisfied.

*Assumption* 1. Assume:

1. The function $\phi \mapsto f(X, \phi)$ is almost surely convex in its second argument and differentiable with Lipschitz continuous gradient, i.e. $\forall \phi_1, \phi_2 \in \Phi$:
$$\mathbb{E}(\|\nabla f(X, \phi_1) - \nabla f(X, \phi_2)\|^2) \leq C\|\phi_1 - \phi_2\|$$
with probability 1 for some $C$.

2. The function $f$ is $\nu$-strongly convex; that is, for all $\phi_1, \phi_2 \in \Phi$:
$$f(X, \phi_1) \geq f(X, \phi_2) + \nabla f(X, \phi_2)^T(\phi_1 - \phi_2)$$
$$+ \frac{\nu}{2}\|\phi_1 - \phi_2\|^2$$

3. There exists $\sigma > 0$ such that $\mathbb{E}[\|\nabla f(X, \phi^*)\|^2) \leq \sigma^2$

4. The function $\phi \mapsto f(X, \phi)$ is almost surely twice differentiable with Lipschitz continuous Hessian $Hf$, i.e. $\forall \phi_1, \phi_2 \in \Phi$:
$$\mathbb{E}(\|(Hf)(X, \phi_1) - (Hf)(X, \phi_2)\|) \leq C'\|\phi_1 - \phi_2\|$$

5. There exists $\tau > 0$ such that $\mathbb{E}[\|\nabla f(X, \phi^*)\|^4] \leq \tau^4$ and there exists a positive definite operator $\Sigma$ such that $\mathbb{E}[\nabla f(X, \phi^*) \otimes \nabla f(X, \phi^*)] \preccurlyeq \Sigma$

6. The function $\mu$ is Lipschitz continuous

It should be noted that, though relatively standard, these assumptions are also quite strong, particularly the assumption of strong convexity of $f$, and may well not hold in practice. In short, the stochastic gradient scheme used in optimizing the bounds may only converge toward a local optimum of the bound $\phi^\dagger$, rather than the global optimum $\phi^*$. When this happens the behavior and rates of convergence will generally be the same, but the error breakdown will become

$$\|\hat{\mu}(d, \phi_K) - \text{EIG}(d)\|_2$$
$$\leq \|\hat{\mu}(d, \phi_K) - \mathcal{B}(d, \phi_K)\|_2 \tag{63a}$$
$$+ \|\mathcal{B}(d, \phi_K) - \mathcal{B}(d, \phi^\dagger)\|_2 \tag{63b}$$
$$+ |\mathcal{B}(d, \phi^\dagger) - \text{EIG}(d)|. \tag{63c}$$

where

$$|\mathcal{B}(d, \phi^\dagger) - \text{EIG}(d)| \geq |\mathcal{B}(d, \phi^*) - \text{EIG}(d)|.$$

We now present our proof for the result, repeating the Theorem itself for convenience.

**Theorem 1.** *Let $\mathcal{X}$ be a measurable space and $\Phi$ be a convex subset of a finite dimensional inner product space. Let $X_1, X_2, ...$ be i.i.d. random variables taking values in $\mathcal{X}$ and $f : \mathcal{X} \times \Phi \to \mathbb{R}$ be a measurable function. Let*

$$\mu(\phi) \triangleq \mathbb{E}[f(X_1, \phi)] \approx \hat{\mu}_N(\phi) \triangleq \frac{1}{N}\sum_{n=1}^{N} f(X_n, \phi)$$

*and suppose that $\sup_{\phi \in \Phi} \|f(X_1, \phi)\|_2 < \infty$. Then $\sup_{\phi \in \Phi} \|\hat{\mu}_N(\phi) - \mu(\phi)\|_2 = \mathcal{O}(N^{-1/2})$. Suppose further that Assumption 1 in Appendix B holds and that $\phi^*$ is the unique minimizer of $\mu$. After $K$ iterations of the Polyak-Ruppert averaged stochastic gradient descent algorithm of [28] with gradient estimator $\nabla_\phi f(X_t, \phi)$, we have $\|\mu(\phi_K) - \mu(\phi^*)\|_2 = \mathcal{O}(K^{-1/2})$ and, combining with the first result,*

$$\|\hat{\mu}_N(\phi_K) - \mu(\phi^*)\|_2 = \mathcal{O}(N^{-1/2} + K^{-1/2}) = \mathcal{O}(T^{-1/2}) \text{ if } N \propto K.$$

**Proof of Theorem 1**

*Proof.* We begin by establishing the uniform convergence of $\hat{\mu}_N(\phi)$ to $\mu(\phi)$, for which we simply use the $L^2$ weak law of large numbers. Specifically, we let $Y_n = f(X_n, \phi)$ and $\varepsilon_N(\phi) = \|\hat{\mu}_N(\phi) - \mu(\phi)\|_2$, then

$$\varepsilon_N^2(\phi) = \mathbb{E}\left(\left[\frac{1}{N}\sum_{n=1}^{N}(Y_n - \mathbb{E}Y_n)\right]^2\right) \tag{64}$$

$$= \mathbb{E}\left(\frac{1}{N^2}\sum_{n=1}^{N}(Y_n - \mathbb{E}Y_n)^2\right) \tag{65}$$

$$= \frac{1}{N^2}\cdot N\text{Var}(Y_n) \tag{66}$$

$$\leq \frac{1}{N}\sup_{\phi\in\Phi}\|f(X_1,\phi)\|_2^2 \tag{67}$$

which is bounded by assumption. Thus

$$\sup_{\phi\in\Phi}\varepsilon_N(\phi) = \mathcal{O}(N^{-1/2}) \tag{68}$$

as required.

We turn now to the stochastic gradient descent convergence. We begin by applying Theorem 3 of [28] using points 1-5 of Assumption 1 to give

$$\|\phi_K - \phi^*\|_2 = \mathcal{O}(K^{-1/2}) \tag{69}$$

and (see [28] page 4)

$$\mathbb{E}\mu(\phi_K) - \mu(\phi^*) = \mathcal{O}(K^{-1/2}). \tag{70}$$

To establish $L^2$ convergence of the function values, it remains to control the variance of $\mu(\phi_K)$. We now invoke point 6 of Assumption 1 to see that, for some constant $B$ (namely the Lipschitz constant for $\mu$),

$$\text{Var}[\mu(\phi_K)] = \mathbb{E}\left[(\mu(\phi_K) - \mathbb{E}\left[\mu(\phi_K)\right])^2\right] \tag{71}$$

$$\leq \mathbb{E}\left[(\mu(\phi_K) - \mu(\mathbb{E}\phi_K))^2\right] \tag{72}$$

$$\leq B^2\mathbb{E}\left[(\phi_t - \mathbb{E}\phi_t)^2\right] \tag{73}$$

$$\leq B^2\|\phi_K - \phi^*\|_2^2. \tag{74}$$

By (69) we conclude $\sqrt{\text{Var}[\mu(\phi_K)]} = \mathcal{O}(K^{-1/2})$. Thus $\mu(\phi_K)$ converges in $L^2$ at the required rate.

Finally, if $\epsilon_K = \|\hat{\mu}_K(\phi_K) - \mu(\phi^*)\|_2$ then

$$\epsilon_K \leq \|\hat{\mu}_K(\phi_K) - \mu(\phi_K)\|_2 + \|\mu_K(\phi_K) - \mu(\phi^*)\|_2$$

$$\leq \|\hat{\mu}_K(\phi_K) - \mu(\phi_K)\|_2 + \sup_{\phi\in\Phi}\|\hat{\mu}_K(\phi) - \mu(\phi)\|_2$$

$$= \mathcal{O}(N^{-1/2} + K^{-1/2})$$

$$= \mathcal{O}(T^{-1/2})$$

as required. $\qquad\square$

Finally, we discuss the necessary extensions for $\mathcal{I}_{\text{m}+\ell}$. The assumptions of the Theorem are subtly different in this case. Specifically, we require Assumption 1 to hold for the integrand of $\mathcal{F}$ rather than the integrand of $\mathcal{I}_{\text{m}+\ell}$, where $\mathcal{F}(d,\phi) = -\mathbb{E}[\log q_m(y|d) + \log q_\ell(y|\theta,d)] + C$ is the loss function that we use to train $\phi$, and require $\mathcal{I}_{\text{m}+\ell}$ to be Lipschitz continuous in $\phi$.

The Monte Carlo error is no different in this setting. However, $\phi^*$ is optimal with respect to $\mathcal{F}(d,\phi)$ rather than $\mathcal{I}_{\text{m}+\ell}$ and the asymptotic bias term is $|\mathcal{I}_{\text{m}+\ell}(d,\phi^*) - \text{EIG}(d)| \leq \mathcal{F}(d,\phi^*)$ by Lemma 2. For the optimization term, we have from equation (69) that $\|\phi_K - \phi^*\|_2 = \mathcal{O}(K^{-1/2})$. Then by the Lipschitz assumption on $\mathcal{I}_{\text{m}+\ell}$, we have $\|\mathcal{I}_{\text{m}+\ell}(d,\phi_k) - \mathcal{I}_{\text{m}+\ell}(d,\phi^*)\|_2 = \mathcal{O}(K^{-1/2})$. The rest of the proof now goes through as above.

## C   Related work

In this section, we provide a more detailed discussion of existing techniques for EIG estimation to complement Sec. 5 in the main text.

One established approach is to use a **Laplace approximation** to the posterior to make fast approximations of EIG [22, 25]

$$\hat{\mu}_{\text{laplace}}(d) \triangleq \frac{1}{N} \sum_{n=1}^{N} \left[ H[p(\theta)] - H[q(\theta|y_n, d)] \right] \tag{75}$$

where $q(\theta|y_n, d)$ is a Laplace approximation to $p(\theta|y_n, d)$ that is computed once for each $y_n \sim p(y|d)$.

Kleinegesse and Gutmann [18] recently suggested an implicit likelihood approach that directly approximates the ratio $r(d, \theta, y) = p(y|\theta, d)/p(y|d)$ using samples from $p(y|\theta, d)$ and $p(y|d)$ and the **Likelihood-Free Inference by Ratio Estimation (LFIRE)** method suggested by [41], which is itself based around logistic regression. This yields the estimator

$$\hat{\mu}_{\text{LFIRE}}(d) \triangleq \frac{1}{N} \sum_{n=1}^{N} \log \hat{r}(d, \theta_n, y_n) \tag{76}$$

where $\log \hat{r}(d, \theta_n, y_n)$ is estimated separately for each pairs of samples $y_n, \theta_n$.

In principal one could also exploit the equivalence between EIG and MI and use other existing MI estimation methods, a number of which were recently summarized by [31]. Of particular note, Belghazi et al. [4] use a bound on MI in the context of generative adversarial neural network training that is based on the **Donsker-Varadhan (DV)** representation of the KL divergence [11]. Specifically, they introduce a parametrized approximation $T(y, \theta|d, \phi)$ to $\log \frac{p(y,\theta|d)}{p(\theta)p(y|d)}$ and then optimize the lower bound

$$\mathcal{L}_{\text{DV}}(d) \triangleq \mathbb{E}_{p(y,\theta|d)}[T(y, \theta|d, \phi)] - \log \left( \mathbb{E}_{p(\theta)p(y|d)}[e^{T(y,\theta|d,\phi)}] \right). \tag{77}$$

The estimator $\hat{\mu}_{\text{DV}}$ is then produced in an analogous manner to $\hat{\mu}_{\text{post}}$.

The EIG has been applied by a number of authors in specific contexts. For instance, the EIG has been used to formulate acquisition functions in Bayesian optimization [16]. More recently, Ma et al. [26] used an EIG-type objective to select features rather than designs for a partial VAE model. The EIG estimation exploits the model structure of the partial VAE. Additionally, and in contrast to this paper, approximations learned using the ELBO are used rather than approximations that are trained using variational objectives that are directly tied to EIG estimation. For further discussion on the implications of using the ELBO (i.e. the reverse KL divergence) in EIG estimation settings, see Appendix G.

As mentioned previously, mutual information bounds are of interest in traditional signal processing [3] and of increasing interest in the deep learning community [31]—although to the best of our knowledge they have not been applied to BOED before. Interestingly, it is lower bounds that are of primary importance in the deep learning setting because of the interplay between MI estimation and the subsequent gradient-based optimization over parameters. This is in contrast to this work, in which we maximize EIG over designs using Bayesian optimization—allowing the use of estimators such as $\hat{\mu}_{\text{m}+\ell}$ that are not, in expectation, bounds.

# D  Experiment details

**Computing** All experiments were run on a machine with 32818560 kB memory, 8 Intel(R) Core(TM) i7-6700 CPU @ 3.40GHz processors, running Fedora 28, Python 3.6.8, Pytorch 1.1.0. To reproduce the results presented in the paper, see `https://github.com/ae-foster/pyro/tree/vboed-reproduce`. The methods in this paper form part of Pyro's OED support, the documentation for which is provided at `http://docs.pyro.ai/en/stable/contrib.oed.html`.

## D.1  EIG estimation accuracy

**A/B test** We consider a classical A/B test, commonly used in marketing and design applications. Here the experiment design is the choice of group sizes: $n$ participants are split between groups A and B of size $n_A$ and $n - n_A$, respectively. For each participant we measure a continuous response $y$. We consider a linear data analysis model

$$\theta \sim N(0, \Sigma_\theta) \qquad y|\theta, d \sim N(X_d\theta, I) \tag{78}$$

where $X_d$ is the $n \times 2$ design matrix with $(1\ 0)$ for the first $n_A$ rows and $(0\ 1)$ for the remainder.

In this example we set the number of participants to be $n = 10$ with 11 designs ($n_A = 0, ..., 10$) and the prior covariance matrix to be

$$\Sigma_\theta = \begin{pmatrix} 10^2 & 0 \\ 0 & 1.82^2 \end{pmatrix} \tag{79}$$

We chose families of variational distributions that include the true posterior (or true marginal). For the amortised posterior, we set $\phi = (A, \Sigma_p)$ with $\phi$ trained separately for each $d$ and let

$$q_p(\theta|y, d, \phi) \sim N(Ay, \Sigma_p) \tag{80}$$

where $A$ is a $10 \times 2$ matrix and $\Sigma_p$ is positive definite. For the marginal, we simply take $\phi = (\mu_m, \Sigma_m)$ and

$$q_m(y|d, \phi) \sim N(\mu_m, \Sigma_m). \tag{81}$$

For NMC and Laplace, no variational families need to be specified.

For LFIRE, we used a parametrization $\phi = (b, \delta, \Lambda)$ and used the ratio estimate

$$\log \hat{r}(y|\theta, d, \phi) = b - (y - \delta)^T \Lambda (y - \delta) \tag{82}$$

where $\Lambda$ is positive definite. This form was chosen to mimic the approximation made by the posterior method, and so reduce the effect of architecture on performance.

For DV, we used a similar critic, namely we set $\phi = (A, \Lambda)$ and

$$T(y, \theta|d, \phi) = -(\theta - Ay)^T \Lambda (\theta - Ay) \tag{83}$$

where $\Lambda$ is positive definite.

The ground truth EIG($d$) was computed analytically. In Table 2, each estimator was allowed 10 seconds computation.

**Preference**    We consider searching for an agent's utility indifference point, using responses that are both *censored* and *corrupted* with non-uniform noise. Let $d \in \mathbb{R}$ and

$$\begin{aligned} \theta &\sim N(\mu_\theta, \sigma_\theta^2) \\ \eta|\theta, d &\sim N(d - \theta, \sigma_\eta^2(1 + |d|)^2) \\ y &= f(\eta) \end{aligned} \tag{84}$$

where

$$f : \mathbb{R} \to [\epsilon, 1 - \epsilon] \tag{85}$$

$$x \mapsto \begin{cases} \epsilon & \text{if } x \leq \text{logit}(\epsilon) \\ 1 - \epsilon & \text{if } x \geq \text{logit}(1 - \epsilon) \\ \frac{1}{1 - e^{-x}} & \text{otherwise} \end{cases} \tag{86}$$

and $\text{logit}(p) = \log p - \log(1 - p)$.

For this example we set $\mu_\theta = -20$, $\sigma_\theta = 20$ and $\sigma_\eta = 1$. We took designs on a linearly spaced grid in $[-80, 80]$. For the variational family for the posterior, we took $\phi = (w, \sigma, \mu_0, \sigma_0, \mu_1, \sigma_1)$ and then

$$q_p(\theta|y, d, \phi) \sim N(\mu_p, \sigma_p^2) \qquad \text{where} \qquad \hat{\eta} = d - \text{logit}(y) \tag{87}$$

$$\mu_p = w\hat{\eta} + (1 - w)\mu_\theta + \mu_0\, \mathbf{1}_{\{y=\epsilon\}} + \mu_1\, \mathbf{1}_{\{y=1-\epsilon\}} \tag{88}$$

$$\sigma_p^2 = \sigma^2 + \sigma_0^2\, \mathbf{1}_{\{y=\epsilon\}} + \sigma_1^2\, \mathbf{1}_{\{y=1-\epsilon\}} \tag{89}$$

For the marginal, we simply took $\phi = (\mu_m, \sigma_m)$ and

$$q_m(y|d, \phi) \sim f \# N(\mu_m, \sigma_m^2). \tag{90}$$

where $\#$ denotes the push-forward measure. We note that this variational family contains the true marginal.

Figure 5: EIG curves for the Preference example, with estimators run until variance is negligible and iterates of $\phi$ are stable to highlight the asymptotic bias.

For LFIRE, we used the parametrization $\phi = (b, b_0, b_1, \delta, \lambda)$ with ratio estimate

$$\hat{\eta} = d - \text{logit}(y) \tag{91}$$

$$\log \hat{r}(y|\theta, d, \phi) = b - \lambda(\hat{\eta} - \delta)^2 + b_0 \, \mathbf{1}_{\{y=\epsilon\}} + b_1 \, \mathbf{1}_{\{y=1-\epsilon\}} \tag{92}$$

For DV, the critic had parametrization $\phi = (b_0, b_1, \delta_i, \delta_0, \delta_1, \lambda_i, \lambda_0, \lambda_1)$ and we set

$$\hat{\eta} = d - \text{logit}(y) \tag{93}$$

$$\lambda = \lambda_i + \lambda_0 \, \mathbf{1}_{\{y=\epsilon\}} + \lambda_1 \, \mathbf{1}_{\{y=1-\epsilon\}} \tag{94}$$

$$\delta = \delta_i + \delta_0 \, \mathbf{1}_{\{y=\epsilon\}} + \delta_1 \, \mathbf{1}_{\{y=1-\epsilon\}} \tag{95}$$

$$T(y, \theta|d, \phi) = -\lambda(\hat{\eta} - \delta)^2 + b_0 \, \mathbf{1}_{\{y=\epsilon\}} + b_1 \, \mathbf{1}_{\{y=1-\epsilon\}} \tag{96}$$

Both these forms were chosen to minimize the differences between the functional forms used for different methods.

The ground truth $\text{EIG}(d)$ was computed by running the marginal method, which is statistically consistent for this example because the true marginal is contained in the variational family, to convergence. The posterior and Laplace methods are both asymptotically biased (see Figure 5) and in this case both make the same (Gaussian) distributional assumption. The posterior method, however, produces better EIG estimates. For the benchmarking results in Table 2, 10 seconds computation was allowed.

**Mixed Effects Regression**   We consider BOED for a mixed effects regression model with a non-linear linking function that will also serve as the basis for the adaptive experiment we run in Sec. 6.3. This class of models is commonly used for analyzing data in a variety of scientific disciplines, where including nuisance variables can be a critical component of the model. In our adaptive experiment, the nuisance variables—i.e. the random effects—are used to account for the variability of individual human participants. Because of the presence of nuisance variables these implicit likelihood models represent a significant challenge for BOED.

We begin by describing the experiment set-up. Participants were presented with a question of the form seen in Figure 6 with the possible images shown in Figure 7. There were two image feature dimensions with 3 levels each. A single image $i$ could therefore be represented as a $1 \times 6$ matrix $X_i$ with two entries 1 and the rest 0. With the left image $i_1$ and right image $i_2$, the question was represented as $X_d = X_{i_1} - X_{i_2}$ encoding the assumed left-right symmetry. We then considered a model for the $i$th participant

$$\theta \sim N(0, \Sigma_\theta) \tag{97}$$

$$\sigma_\psi^{-2} \sim \Gamma(\alpha_\psi, \beta_\psi) \tag{98}$$

$$\psi_i|\sigma_\psi \sim N(0, \sigma_\psi^2 I_6) \tag{99}$$

$$\sigma_k^{-2} \sim \Gamma(\alpha_k, \beta_k) \tag{100}$$

$$\log k_i|\sigma_k \sim N(0, \sigma_k^2) \tag{101}$$

$$\eta|\theta, \psi_i, k_i, d \sim N(k_i(X_d\theta + X_d\psi_i), \sigma_\eta^2) \tag{102}$$

$$y = f(\eta) \tag{103}$$

where $f$ is the censored sigmoid defined in (86) and $i \in \{1, ..., 8\}$ as there were 8 different participants.

The actual prior values of the parameters used were

$$\Sigma_\theta = 100 I_6 \qquad \sigma_\eta = 10 \tag{104}$$
$$\alpha_\psi = \beta_\psi = \alpha_k = \beta_k = 2 \tag{105}$$

We begin by discussing the variational families used to estimate the EIG.

For the posterior estimator of EIG, we took $\phi = (A, \Sigma_\mathrm{p})$ and

$$\hat{\eta} = \mathrm{logit}(y) \tag{106}$$
$$q_p(\theta|y, d, \phi) \sim N(A\hat{\eta}, \Sigma_\mathrm{p}) \tag{107}$$

For the marginal + likelihood estimator, we set $\phi = (\mu_\mathrm{m}, \sigma_\mathrm{m}, \mu_\ell, \sigma_\ell, \xi)$ and took

$$q_m(y|d, \phi) \sim f\#N(\mu_\mathrm{m}, \sigma_\mathrm{m}^2) \tag{108}$$
$$q_\ell(y|\theta, d, \phi) \sim f\#N(e^\xi X_d \theta + \mu_\ell, \sigma_\ell^2) \tag{109}$$

For LFIRE, we used $\phi = (b, \delta, \lambda)$ and then took

$$\hat{\eta} = \mathrm{logit}(y) \tag{110}$$
$$\log \hat{r}(y|\theta, d, \phi) = b - \lambda(\hat{\eta} - \delta)^2 \tag{111}$$

For DV, we used $\phi = (\lambda, \xi)$ and

$$\hat{\eta} = \mathrm{logit}(y) \tag{112}$$
$$T(y, \theta|d, \phi) = -\lambda(\hat{\eta} - e^\xi X_d \theta)^2 \tag{113}$$

For benchmarking, we computed the ground truth using a variant of NMC. Specifically, we note that

$$p(y|d) = \mathbb{E}_{p(\theta, \psi, k)}[p(y|\theta, \psi, k, d)] \tag{114}$$
$$p(y|\theta, d) = \mathbb{E}_{p(\psi, k)}[p(y|\theta, \psi, k, d)] \tag{115}$$

and for this model, we can sample directly from $p(\psi, k)$. These identities allow us to estimate the marginal and likelihood by Monte Carlo, and then combine in a NMC estimator for $\mathrm{EIG}(d)$. Whilst inefficient, this estimator is statistically consistent.

We allowed 60 seconds computation per estimator to compute the results of Table 2. Encouragingly, we find that our variational estimators outperform the LFIRE and DV baselines on this model and exhibit low errors even though they both make suboptimal distributional assumptions about the posterior/marginal.

**Extrapolation** We consider designing experiments to reduce posterior uncertainty in the model prediction at another point in design space—a point that we cannot experiment on directly. For this example, we take $\psi \sim N(\mu_\psi, \Sigma_\psi)$ and

$$\theta|\psi \sim \mathrm{Bernoulli}(\mathrm{logit}^{-1}(X_\theta \psi))$$
$$y|\psi, d \sim \mathrm{Bernoulli}(\mathrm{logit}^{-1}(X_d \psi))$$

where $X_\theta = \begin{pmatrix} 1 & -\frac{1}{2} \end{pmatrix}$ and $X_d = \begin{pmatrix} -1 & d \end{pmatrix}$ for $d \in \mathbb{R}$. Interestingly, this model admits efficient sampling of $y, \theta \sim p(y, \theta|d)$ but *not* $y \sim p(y|\theta, d)$. Therefore, whilst the posterior, marginal + likelihood and DV methods are all applicable, LFIRE is not.

For the posterior method we set $\phi = (l_0, l_1)$ and

$$l_\mathrm{p}(y) = l_1 y + l_0(1 - y) \tag{116}$$
$$q_p(\theta|y, d, \phi) \sim \mathrm{Bernoulli}(\mathrm{logit}^{-1}(l_\mathrm{p}(y))). \tag{117}$$

We computed the prior entropy, which is not analytically tractable here, using a MC estimator, noting that $\theta$ has a finite sample space.

For the marginal + likelihood method, we let $\phi = (l, l_0, l_1)$ and then

$$q_m(y|d, \phi) \sim \text{Bernoulli}(\text{logit}^{-1}(l)) \tag{118}$$

$$l_\ell(\theta) = l_1\theta + l_0(1 - \theta) \tag{119}$$

$$q_\ell(y|\theta, d, \phi) \sim \text{Bernoulli}(\text{logit}^{-1}(l_\ell(\theta))). \tag{120}$$

Finally, for DV, we let $\phi = (w_y, w_\theta, w_{y\theta})$ and took

$$T(\theta, y|d, \phi) = w_y y + w_\theta \theta + w_{y\theta} y\theta. \tag{121}$$

The ground truth EIG was computed using MC, noting that the sample spaces for $y, \theta$ are finite in this example. 10 seconds computation per methods was allowed for the results in Table 2.

## D.2   End-to-end sequential experiments

**Mechanical Turk experiment**   We begin by describing the experiment itself. Participants were presented with a question of the form seen in Figure 6 with the possible images shown in Figure 7. There were two image feature dimensions with 3 levels each. A single image $i$ could therefore be represented as a $1 \times 6$ matrix $X_i$ with two entries 1 and the rest 0. With the left image $i_1$ and right image $i_2$, the question was represented as $X_d = X_{i_1} - X_{i_2}$ encoding the assumed left-right symmetry.

The model and EIG estimation were the same as the mixed effects model in Sec. D.1. When optimizing the EIG to select designs $d_t$, we estimated EIG across all candidate designs. We allowed a 30s turnaround to learn the posterior from the previous data, estimate the EIG, select the next design, and present it to the user. We estimated the EIG in parallel for all 36 designs to select the best design at each step. For each independent run of the experiment there were 8 participants, each answering 10 questions. This allowed the interplay between fixed effects and random effects to be apparent.

Because we used this model to run an adaptive experiment, we required a variational family to learn the full posterior (over random effects and hyperparameters as well as $\theta$).

For the full variational inference of the posterior used when we receive actual data, we used a partial mean-field approximation. Specifically, we set $q(\theta, \sigma_\psi, (\psi_i)_{i=1}^8, \sigma_k, (k_i)_{i=1}^8)$ to be

$$\theta \sim N(\mu_\theta, \Sigma_\theta) \tag{122}$$

$$\sigma_\psi^{-2} \sim \Gamma(\alpha_\psi, \beta_\psi) \tag{123}$$

$$\psi_i|\theta \sim N(A(\theta - \mu_\theta) + \mu_{\psi_i}, \Sigma_{\psi_i}) \tag{124}$$

$$\sigma_k^{-2} \sim \Gamma(\alpha_k, \beta_k) \tag{125}$$

$$\log k_i \sim N(\mu_{k_i}, \sigma_{k_i}^2) \tag{126}$$

and we learned the variational parameters $\mu_\theta, \Sigma_\theta, \alpha_\psi, \beta_\psi, A, \mu_{\psi_i}, \Sigma_{\psi_i}, \alpha_k, \beta_k, \mu_{k_i}, \sigma_{k_i}$ by conventional (not amortized) variational inference. Note that, under this approximate posterior, $\theta$ is multivariate Gaussian so we can compute its entropy analytically.

Finally we ran an additional experiment identical to the first, but using simulated data rather than human responses. We took

$$\theta = \begin{pmatrix} -30 & 30 & 0 & -12 & -6 & 18 \end{pmatrix}. \tag{127}$$

We simulated the random effects $\psi, k$ from the prior and used the prior value $\sigma_\eta = 10$. The entropy results are presented in Figure 8. As expected, BOED decreases posterior uncertainty more quickly.

## D.3   Constant Elasticity of Substitution (CES) experiment

We begin by describing the experiment set-up. The economic agent is presented with a sequence of designs $d$. Each designs comprises two baskets $\mathbf{x}$ and $\mathbf{x}'$ of goods. The agent then indicates which basket they prefer on a one-dimensional slider—they may indicate a strong preference, weak preference, or indifference.

Figure 6: A screenshot of the question answering interface used by human participants in the adaptive experiment in Sec. 6.3.

Figure 7: The nine characters we used in the adaptive experiment in Sec. 6.3. They vary along two feature dimensions: the mouth (smile, frown, showing teeth) and eyebrows.

Figure 8: Evolution of the posterior entropy of the fixed effects in the Mechanical Turk experiment in Sec. 6.3 with simulated data. We depict the mean and $\pm 1$ std. err. from 10 experimental trials.

To model the agent's responses, we use the CES utility model [2] which defines a utility

$$U(\mathbf{x}) = \left( \sum_i x_i^\rho \alpha_i \right)^{1/\rho} \tag{128}$$

for a basket of goods $\mathbf{x}$. In this experiment, we took baskets $\mathbf{x} \in [0, 100]^3$ representing non-negative quantities of three commodities.

Extending the preference example in the previous section, we assume the agent, when asked to compare baskets $\mathbf{x}$ and $\mathbf{x}'$ and indicate their preference on a slider, base their response on $U(\mathbf{x}) - U(\mathbf{x}')$. Specifically, we use the following likelihood model

$$\rho \sim \text{Beta}(a_\rho, b_\rho) \tag{129}$$
$$\boldsymbol{\alpha} \sim \text{Dirichlet}(\boldsymbol{c_\alpha}) \tag{130}$$
$$\log u \sim N(\mu_u, \sigma_u^2) \tag{131}$$
$$\eta|\rho, \boldsymbol{\alpha}, d \sim N(u \cdot (U(\mathbf{x}) - U(\mathbf{x}')), \sigma_\eta^2 u^2 (1 + \|\mathbf{x} - \mathbf{x}'\|)^2) \tag{132}$$
$$y = f(\eta) \tag{133}$$

This represents a challenging experiment design problem for a number of reasons. First, for large values of $U(\mathbf{x}) - U(\mathbf{x}')$ the agent's response will be predictable gaining little information. For very different baskets ($\|\mathbf{x} - \mathbf{x}'\|$ large) the responses will be noisy indicating our intuition that it is more difficult to compare very different baskets. However, very similar baskets will have similar utilities and the agent will be predictably indifferent. Optimal designs therefore lie in a sweet spot where: i) baskets are similar to avoid high noise regions, but dissimilar enough to be informative; and ii) the difference in utility is close to 0 under the current posterior. BOED is able to trade off these considerations in a principled manner.

For this specific example we took

$$a_\rho = b_\rho = 1 \qquad \boldsymbol{c_\alpha} = (1, 1, 1) \tag{134}$$
$$\mu_u = 1 \qquad \sigma_u = 3 \tag{135}$$
$$\sigma_\eta = 0.005 \tag{136}$$

To estimate the EIG, we used a marginal guide based on the one used in the preference example. Specifically, we set $\phi = (\mu_\text{m}, \sigma_\text{m}, p_0, p_1)$ and

$$r(y|d, \phi) \sim f \# N(\mu_\text{m}, \sigma_\text{m}^2), \tag{137}$$

$$q_p(y|d, \phi) = \begin{cases} \epsilon & \text{with probability } p_0 \\ 1 - \epsilon & \text{with probability } p_1 \\ r(y|d, \phi) & \text{with probability } 1 - p_0 - p_1 \end{cases} \tag{138}$$

where $\#$ denotes the push-forward measure. This is simply a mixture of a discrete distribution on end-points with a sigmoid transformed Gaussian.

To select designs, we used Bayesian optimization with a Matern52 kernel with lengthscale 20 and variance set empirically. Both $\hat{\mu}_\text{marg}$ and $\hat{\mu}_\text{NMC}$ were allowed the same time budget to select designs and used an identical Bayesian optimization procedure. Random designs were chosen uniformly on $[0, 100]^6$.

To learn the posterior at subsequent steps we used a mean-field variational approximation with the same families as the prior. That is, we updated the parameters $a_\rho, b_\rho, \boldsymbol{c_\alpha}, \mu_u, \sigma_u$ and left the structure otherwise intact. The RMSEs of Figure 4 were expectations over the posterior: $\left( \mathbb{E}_{p(\theta|d_{1:t}, y_{1:t})} [\|\theta - \theta^*\|^2] \right)^{1/2}$.

# E   Additional experiments

## E.1   Death process

We examine experimental design for the simple continuous time process considered in [9] and [18], arising in epidemiology. Consider a population with fixed size $N$ that is initially healthy at time

(a) Exact EIG     (b) Posterior LogNormal     (c) Truncated Normal     (d) Laplace

Figure 9: EIG surfaces estimated by four methods for the two-dimensional design $(t_1, t_2)$ for the continuous time model described in Sec. E.1. The optimal design $(t_1^*, t_2^*)$ determined by each method is indicated with a cross. The posterior method with a LogNormal variational distribution yields nearly exact results. The posterior method with a Truncated Normal distribution and the Laplace method are not as accurate but still result in designs with large EIG. Note that the EIG has been scaled for interpretability and that all four figures use a common scale. The errors of these estimators are examined more closely in Figure 10.

(a) Posterior LogNormal     (b) Posterior Truncated Normal     (c) Laplace

Figure 10: Absolute EIG errors corresponding to the estimates depicted in Fig. 9. The optimal design $(t_1^*, t_2^*)$ determined by an exact method is indicated with a star. The absolute error of the LogNormal Posterior estimate is $\sim 10^{-3}$ across the design space. The mean absolute error of the Laplace EIG estimates across the design space is about 30% higher than for the Posterior method with a Truncated Normal variational distribution. In this case the Laplace method results in an upper bound, while (as always) both Posterior methods yield a lower bound. All three figures have the same scale as Fig. 9, except for the LogNormal errors, which have been scaled by an additional factor of 100.

$t = 0$, with individuals becoming infected at a constant rate $b$ as time evolves. We consider a design space $d = (t_1, t_2)$, where $0 \leq t_1 \leq t_2$, corresponding to the times at which we measure the number of infected individuals. We place a log-normal prior on the infection rate $b$.

For this example, we investigate how the choice of variational family affects the asymptotic bias. In Fig. 9 we compare the EIG surfaces obtained using four estimators: i) an exact method that uses brute force quadrature; ii) $\hat{\mu}_{\text{post}}$ with a log-normal variational distribution; iii) $\hat{\mu}_{\text{post}}$ with a truncated normal variational distribution; and iv) the Laplace approximation $\hat{\mu}_{\text{laplace}}$. The log-normal family matches the true posterior best, giving mean absolute errors $\sim 10^{-3}$. The second posterior method and the Laplace approximation both make the same distributional assumption, but Laplace results in absolute errors that are about 30% higher than for the posterior method. See Fig. 10 for a closer analysis of the errors of the approximate methods.

**Experimental details**    The likelihood for observing $(I_1, I_2)$ infected individuals from a population of size $N$ at times $(t_1, t_2)$ is given by [12]:

$$p(I_1, I_2 | b, t_1, t_2) = \frac{N!}{I_1!(I_2 - I_1)!(N - I_2)!} \left[1 - e^{-bt_1}\right]^{I_1} \times$$

$$\left[1 - e^{-b(t_2 - t_1)}\right]^{I_2 - I_1} \left[e^{-bt_1}\right]^{I_2 - I_1} \left[e^{-bt_2}\right]^{N - I_2} \tag{139}$$

The prior over the infection rate $b > 0$ is taken to be

$$\log b \sim N(\mu_b, \sigma_b) \tag{140}$$

so that the joint density is given by

$$p(I_1, I_2, b | t_1, t_2) = p(I_1, I_2 | b, t_1, t_2) p(b) \tag{141}$$

In our experiment we choose $N = 10$, $\mu_b = 0$, and $\sigma_b = 0.25$. The figures are scaled such that the maximum EIG over the design space (as computed with the exact method) is 1.0. For all four EIG estimation methods we use quadrature and exact summation over the outcomes $(I_1, I_2)$ where appropriate to obtain maximally accurate results. That is, the obtained results are only constrained by the methods themselves and not the computational budget used. Note that we do not make use of any kind of amortization.

## F  Consistent EIG estimation with control variates

In this section, we show that an approximation to the marginal density $q_m(y|d)$ can be used a control variate. Control variates are a means to reduce the variance of Monte Carlo estimators by using expectations which can be computed analytically. Here, we assume that, for every $\theta$, the KL divergence $\mathrm{KL}\,(\,p(y|\theta, d)\,||\,q_m(y|d)\,)$ can be computed analytically. For example, this would be the case if both $p(y|\theta, d)$ and $q_m(y|d)$ were Gaussian.

We begin by writing the EIG as

$$\mathrm{EIG}(d) = \mathbb{E}_{p(y, \theta | d)} \left[ \log \frac{p(y|\theta, d)}{p(y|d)} \right] \tag{142}$$

$$= \mathbb{E}_{p(y, \theta | d)} \left[ \log \frac{p(y|\theta, d)}{q_m(y|d)} \right] + \mathbb{E}_{p(y, \theta | d)} \left[ \log \frac{q_m(y|d)}{p(y|d)} \right] \tag{143}$$

$$= \mathbb{E}_{p(\theta)} \left[ \mathrm{KL}\,(\,p(y|\theta, d)\,||\,q_m(y|d)\,) \right] - \mathrm{KL}\,(\,p(y|d)\,||\,q_m(y|d)\,). \tag{144}$$

We can now use our assumption on the first term,

$$\mathbb{E}_{p(\theta)} \left[ \mathrm{KL}\,(\,p(y|\theta, d)\,||\,q_m(y|d)\,) \right] \to \mathbb{E}_{p(\theta)} \left[ \text{analytic function of } \theta \right] \tag{145}$$

and this expectation can be computed efficiently with conventional Monte Carlo. For the second term, we use Nested Monte Carlo

$$\mathrm{KL}\,(\,p(y|d)\,||\,q_m(y|d)\,) \approx \frac{1}{N} \sum_{n=1}^{N} \log \frac{\frac{1}{M} \sum_{m=1}^{M} p(y_n | \theta_m, d)}{q_m(y_n | d)} \tag{146}$$

where $y_n \overset{\text{i.i.d.}}{\sim} p(y|d)$ and $\theta_m \overset{\text{i.i.d.}}{\sim} p(\theta)$. The key benefit of this approach is that this estimator may have lower variance than a direct NMC estimator of $\mathrm{EIG}(d)$. Indeed, if we let $A = \log \left( \frac{1}{M} \sum_{m=1}^{M} (y | \theta_m, d) \right)$ and $B = \log q_m(y|d)$ then the variance of the estimator in (146) is

$$\mathrm{Var}(A - B) = \mathrm{Var}(A) + \mathrm{Var}(B) - 2\,\mathrm{Cov}(A, B) \tag{147}$$

so the variance will be low when $\mathrm{Cov}(A, B)$ is large. We can expect this to happen when $q_m(y|d)$ is a good approximation to the true marginal density $p(y|d)$.

Finally, note that just like $\hat{\mu}_{\mathrm{VNMC}}$, this estimator is consistent, i.e. it will converge to the EIG as $N, M \to \infty$.

## G  KL $(\,q\,||\,p\,)$ versus KL $(\,p\,||\,q\,)$

In Appendix A.1, we showed that our posterior estimator is implicitly minimizing the following expected KL divergence

$$\mathrm{EIG}(d) - \mathcal{L}_{\mathrm{post}}(d) = \mathbb{E}_{p(y|d)} \left[ \mathrm{KL}\,(\,p(\theta|y, d)\,||\,q_p(\theta|y, d)\,) \right]. \tag{148}$$

In variational inference, the inner KL divergence is referred to as the *forward* KL. In this section, we compare our approach with a similar approach which also uses a posterior approximation, but instead minimize the *reverse* KL divergence, $\mathrm{KL}\,(\,q_p(\theta|y, d)\,||\,p(\theta|y, d)\,)$.

Specifically, we explore how the reverse KL divergence exhibits discontinuous behaviour that could be problematic in the context of EIG estimation. We begin by writing the posterior estimator as

$$\mathcal{L}_{\mathrm{post}}(d) = \mathbb{E}_{p(y|d)} \left[ \mathbb{E}_{p(\theta|y)} [\log q_p(\theta|y, d)] \right] + H[p(\theta)]. \tag{149}$$

(a) Forward and reverse KL minimization      (b) Discontinuity in the partial KL

Figure 11: (a) Normal variational distributions found by fitting to a target posterior that is a mixture with two distinct Normal components. In both plots, the target posterior is a mixture of $N(\mu_1, 0.5^2)$ and $N(\mu_2, 1.0^2)$ and we vary $\Delta\mu_{\mathrm{post}} = \mu_1 - \mu_2$. In the top plot, the gap between the two components is $\Delta\mu_{\mathrm{post}} = 3.0$, while in the bottom plot $\Delta\mu_{\mathrm{post}} = 3.3$. In contrast to the behaviour resulting from forward KL minimization, the mode-seeking behaviour of reverse KL minimization leads to a large change in the corresponding optimal variational distribution from top to bottom. (b) We plot the partial KL as we vary $\Delta\mu_{\mathrm{post}}$ for the target posterior described in (a). The partial KL as estimated by reverse KL minimization exhibits a sharp discontinuity as the gap between the two components crosses $\Delta\mu_{\mathrm{post}} \approx 3.18$.

The term involving $q_p$ is the expectation of the partial KL, $\mathbb{E}_{p(\theta|y)}[\log q_p(\theta|y, d)]$. We will show that reverse KL minimization can lead to a discontinuity in the partial KL.

We consider two possible methods for choosing $q_p$. We know from (148) that the optimal choice of $q_p$ within a variational family $\mathcal{Q}$ is

$$q_{\mathrm{forward}}(\theta|y, d) \triangleq \underset{q \in \mathcal{Q}}{\arg\min} \, \mathrm{KL}\left( \, p(\theta|y, d) \, \| \, q(\theta) \, \right). \tag{150}$$

An alternative choice is

$$q_{\mathrm{reverse}}(\theta|y, d) \triangleq \underset{q \in \mathcal{Q}}{\arg\min} \, \mathrm{KL}\left( \, q(\theta) \, \| \, p(\theta|y, d) \, \right) \tag{151}$$

which is the form usually seen in variational inference. The posterior method outlined in Section 3 attempts to learn $q_{\mathrm{forward}}$ for each $y$ by maximizing the bound $\mathcal{L}_{\mathrm{post}}$. In this appendix, we show that the alternative $q_{\mathrm{reverse}}$, as well as resulting in less accurate EIG estimates in light of (148), can lead to discontinuities in the partial KL.

Minimizing the reverse KL can result in the well-known behaviour of mode-locking—and thus mode-dropping—which in our context can result in significant misestimates of the EIG. Furthermore, since this mode-locking behaviour is discontinuous (so that it can occur for a particular design $d$ but not for a neighbouring design $d'$) it can potentially result in large design-dependent bias in EIG estimation. For a quantitative exploration of this phenomenon for two bimodal posteriors and a Normal family of variational distributions $\mathcal{Q}$ see Figure 11.