[Reviews · NeurIPS 2019]

Reviewer 1



I have read the author response and am keeping my scores as is. I didn't have any real concerns so the response was mainly aimed toward other reviewers. The paper should probably get an oral presentation. *** I think this is a good application of the ideas of inference networks (finding a good use case for the functional approximation of a difficult distribution of interest). It is highly impactful in experimental design, of course, and noticing how the functional approximation could be taken advantage of in the context of nested MC in particular is not necessarily straightforward to spot. The paper is well written and communicated. It is very easy to follow and covers a vast breadth of material succinctly yet still thoroughly. With that said, it may be worth considering placing some concrete examples along the way. For example, early on in Sec 2 you could have followed your psychology motivation and provided what the design d looks like, the data y, the model p(y, \theta | d) and even what sampling y means intuitively in this case.

Reviewer 2



Quality: Is the submission technically sound? Are claims well supported by theoretical analysis or experimental results? Is this a complete piece of work or work in progress? Are the authors careful and honest about evaluating both the strengths and weaknesses of their work? Overall, the paper seems mathematically sound. Two of the four variational estimators introduced are already known in the context of approximating mutual information. Hence, results for these follow easily from the connection between mutual information and expected information gain and the original paper on variational information maximization. The bound property of the third estimator and the convergence rate analysis are based on established techniques. Clarity: Is the submission clearly written? Is it well organized? (If not, please make constructive suggestions for improving its clarity.) Does it adequately inform the reader? (Note: a superbly written paper provides enough information for an expert reader to reproduce its results.) The paper is well-written and self-contained. The problem is well motivated. Theoretical results are introduced on an intuitive level with the main ideas explained in the text and the mathematical details moved to the appendix. Originality: Are the tasks or methods new? Is the work a novel combination of well-known techniques? Is it clear how this work differs from previous contributions? Is related work adequately cited? (Abstracts and links to many previous NeurIPS papers are available here.) The work applies variational estimators of mutual information to optimal experimental design. This is a new take on the classical field of BOED and conceptionally interesting. In a second step, the authors combine their ideas with standard nested MC estimation. This allows to trade speed for accuracy and can be important for practitioners. Significance: The key idea of combining variational inference and BOED is helpful in several ways. Most importantly, the approach has the potential to be included in probabilistic programming toolboxes which would help to make BOED more accessible to practitioners without statistical background. In addition, the variational approach may inspire future work on more complex problems for which traditional approximations such as Laplace are not well-suited. Additional comments: Section 3, variational marginal: The Barber-Agakov paper is stated as a reference. However, this paper seems to deal only with the lower bound which corresponds to the variational posterior. Eq. 9: The VNMC-estimator section could benefit from some additional high-level information. It is not clear from the exposition, where the equation comes from or why one should consider replacing the usual expectation for computing EIG with the extended form. Related Work: A recent publication (ref 31) deals with variational bounds on mutual information but is only mentioned in the appendix. Since both topic and time of publication are so close, this work should be discussed in the main text. Table 1: A reader might wonder here about the baseline methods because they are introduced later in the text. I suggest adding a quick reference such as “Baselines explained in Section 5”. App. A1 and A2: These proofs are unnecessary because the bounds are known. A reference would be sufficient. At least a remark indicating that the proofs are “provided for completeness” or similar is appropriate. App. A3: The proof refers to U_marg several times. Is this supposed to be U_VNMC?

Reviewer 3



This paper provides four different estimators for expected information gain in a Bayesian optimal experimental design framework, with the objective of using amortized variational inference to reduce the computational cost. The main idea is using variation approximations with shared parameters for either posterior of parameters of interest, or the marginal of outcome given the design. Furthermore, an importance sampling estimator with asymptotic consistency. Over the idea is interesting and its presentation is neat. Theoretical study of convergence for the proposed estimators and their performance in practice is also provided. Some minor comments: 1. In the sequential setting, it seems that the implied assumption is independence of designs across different times, as entropy of the prior for parameters is assumed constant w.r.t design [line 169]. Does this hold in practice? 2. Regarding the performance results in Table 2, somehow the lower bias of \mu_{m+l} compared to \mu_{post} seems counter-intuitive, as the former uses two variational approximations. Are parameters shared between q_m and q_l, so that biases cancel out?

[Author Response · NeurIPS 2019]

We would like to thank all the reviewers for their thoughtful reviews and helpful suggestions. We were delighted to
see that the paper was unanimously well received and were particularly happy to see that the reviewers agreed that the
work has the potential to make a big impact. We are excited about variational BOED and follow-up work indicates that
VBOED opens the door to further developments in machine learning, statistics and other fields. We turn now to specific
comments and questions.

**Reviewer 1**   Thank you for your review.

1. More concrete examples in Section 2 is a great suggestion which we will implement in time for the camera
ready, if accepted. To be specific, in the psychology trial example, the design $d$ is the choice of question, $\theta$
represents the parameters of an underlying psychological model $p(y|\theta, d)$, and $y$ is the participant's response.

**Reviewer 2**   Thank you for your review.

1. Thank you for pointing out our mistake with the reference for the variational marginal bound. We will be sure
to correct this.

2. We are glad you brought the issue of high-level intuition for VNMC to our attention and we will make updates
to be clearer here. To give some additional explanation, both the NMC and VNMC estimators take the form

$$\text{EIG}(d) \approx \frac{1}{N} \sum_{n=1}^{N} \log \frac{p(y_n|\theta_n, d)}{\hat{p}(y_n|d)} \text{ where } y_n, \theta_n \overset{\text{i.i.d.}}{\sim} p(\theta)p(y|\theta, d) \tag{1}$$

where, for NMC, $\hat{p}_{\text{NMC}}(y|d) = \frac{1}{M} \sum_{m=1}^{M} p(y|\theta_m, d)$ where $\theta_m \overset{\text{i.i.d.}}{\sim} p(\theta)$. Written in this way, we see that
NMC is approximating $p(y|d)$ using samples from $p(\theta)$. We expect better approximations of $p(y|d)$ using
samples from a proposal $q_v(\theta|y)$ that is close to the posterior $p(\theta|y, d)$, i.e.

$$\hat{p}_{\text{VNMC}}(y|d) = \frac{1}{M} \sum_{m=1}^{M} \frac{p(\theta)p(y|\theta_m, d)}{q_v(\theta|y)} \text{ where } \theta_m \overset{\text{i.i.d.}}{\sim} q_v(\theta|y) \tag{2}$$

which leads to the VNMC estimator. It is also important to establish the bounds of Lemma 1, because these
allow a variational training method for $q_v$.

3. We agree that Poole, et al. (2019) is an important reference and will make sure it is discussed in the main text.

4. Table 1: thanks for picking this up. We agree a pointer to Section 5 would be helpful.

5. A1 and A2: we will be sure to indicate that these proofs were added for completeness and add references.

6. A3: thanks for picking up these typos!

**Reviewer 3**   Thank you for your review.

1. This is an interesting point where we could have been clearer. In the sequential setting, we assume that

$$p(y_{1:t}, \theta|d_{1:t}) = p(\theta) \prod_{\tau=1}^{t} p(y_\tau|\theta, d_\tau) \tag{3}$$

which says that experiments are conditionally independent given designs and $\theta$. After conducting experiments
$1, ..., t-1$ we have $p(y_t, \theta|y_{1:t-1}, d_{1:t-1}, d_t) = p(\theta|y_{1:t-1}, d_{1:t-1})p(y_t|\theta, d_t)$ and now select $d_t$ conditional
on $d_{1:t-1}, y_{1:t-1}$ using the new prior $p(\theta|d_{1:t-1}, y_{1:t-1})$. The entropy of this new prior distribution is a
constant with respect to $d_t$ which is why we can drop it on line 169. The new prior still makes its presence felt
in the other term in $\mathcal{L}_{\text{post}}$, namely $\mathbb{E}_{p(\theta|d_{1:t-1}, y_{1:t-1})p(y|\theta, d_t)}[\log q_p(\theta|y, d_t)]$.

2. We agree that the lower bias of $\mu_{m+\ell}$ compared to $\mu_{\text{post}}$ may at first sight be unintuitive. Although $\mu_{m+\ell}$ uses
two variational approximations compared to one for $\mu_{\text{post}}$, the approximations are for variables which have
different dimensionality. If $y$ has a lower dimension than $\theta$, it may make sense to use $\mu_{m+\ell}$ instead of $\mu_{\text{post}}$.
On the other hand, $\mu_{m+\ell}$ uses the same approximation as $\mu_{\text{marg}}$ plus an extra one. We would never recommend
using $\mu_{m+\ell}$ in place of $\mu_{\text{marg}}$ (cf. line 186), but one may have to fall back on $\mu_{m+\ell}$ in an implicit likelihood
setting.
In our experiments, parameters were not shared between $q_m$ and $q_\ell$, although this is an interesting idea that
could further reduce the bias. A limited discussion of this idea appears at the end of Section A.4 (it can actually
lead to a new lower bound, but requires additional assumptions).

# References

Poole, Ben, et al. "On variational bounds of mutual information." arXiv preprint arXiv:1905.06922 (2019).


[Meta-Review · NeurIPS 2019]

This paper provides a novel, sound, and seemingly practical method for Bayesian optimal experimental design. The basic idea is to create upper and lower bounds on the information gain analogously to existing ideas to upper and lower-bound the marginal likelihood in traditional VI. This is an important problem, and the reviewers were unanimous that the ideas had merit.